# Benthic diel oxygen variability and stress as potential drivers for animal diversification in the Neoproterozoic-Palaeozoic

Emma U. Hammarlund [1] ✉, Anuraag Bukkuri[1,2,3], Magnus D. Norling[4], Mazharul Islam [1], Nicole R. Posth [5], Etienne Baratchart[1], Christopher Carroll[1], Sarah R. Amend [6], Robert A. Gatenby [7], Kenneth J. Pienta [6], Joel S. Brown[7], Shanan E. Peters [8] & Kasper Hancke [4]

The delay between the origin of animals in the Neoproterozoic and their Cambrian diversification remains perplexing. Animal diversification mirrors an expansion in marine shelf area under a greenhouse climate, though the extent to which these environmental conditions directly influenced physiology and early organismal ecology remains unclear. Here, we use a biogeochemical model to quantify oxygen dynamics at the sunlit sediment-water interface over day-night (diel) cycles at warm and cold conditions. We find that warm temperatures dictated physiologically stressful diel benthic oxic-anoxic shifts over a nutrient-rich shelf. Under these conditions, a population-and-phenotype model further show that the benefits of efficient cellular oxygen sensing that can offer adaptations to stress outweigh its cost. Since diurnal benthic redox variability would have expanded as continents were flooded in the end-Neoproterozoic and early Palaeozoic, we propose that a combination of physiological stress and ample resources in the benthic environment may have impacted the adaptive radiation of animals tolerant to oxygen fluctuations.

Animals diversified dramatically during the first half of the Cambrian Period (540–490 million years ago; Ma)[1] in the shallow benthic environment[2] (Fig. 1A). Multiple studies have investigated how temperature, nutrients and oxygen gradients influenced animal diversity in the Cambrian Period[3]. However, the specific mechanisms by which these abiotic conditions influenced subsequent changes in animal diversity patterns at the Neoproterozic-Paleozoic boundary remains unresolved, and no single hypothesis fully explains the timing of the event and nor is it likely to[4]. For example, it is suggested that environmental oxygen concentrations increased above a threshold permissive for a diversity of animal species at some point in time, but we

lack understanding of a line of events that would have spurred the diversification of animals after this threshold[5]. In contrast to how other biotic turnover events are linked to harsh conditions[6], the Cambrian explosion is rarely explored as the result of physiological stress. Recent investigations, however, note that recurring marine anoxia and heterogenous water column oxygenation may have accelerated evolutionary innovations at the Neoproterozic-Paleozoic boundary[7–9] and over the Phanerozoic Eon[6,9]. More generally, it is well established that cellular and physiological stress is a driver for change in evolutionary biology[10]. Rapid evolutionary change in organisms as diverse as Ordovician trilobites and modern fruit flies is associated with

[1]Tissue Development and Evolution (TiDE) Group, Department of Experimental Medical Science, Lund University, Lund, Sweden. [2]Department of Computational and Systems Biology, University of Pittsburgh, Pittsburgh, PA, USA. [3]Center for Evolutionary Biology and Medicine, University of Pittsburgh, Pittsburgh, PA, USA. [4]Norwegian Institute for Water Research (NIVA), Oslo, Norway. [5]Department of Geosciences and Natural Resource Management (IGN), Geology Section, University of Copenhagen, Copenhagen, Denmark. [6]The Cancer Ecology Center, Brady Urological Institute, Johns Hopkins School of Medicine, Baltimore, MD, USA. [7]Department of Integrated Mathematical Oncology, Moffitt Cancer Center, Tampa, FL, USA. [8]Department of Geoscience, University of Wisconsin–Madison, Madison, WI, USA. ✉e-mail: emma.hammarlund@med.lu.se

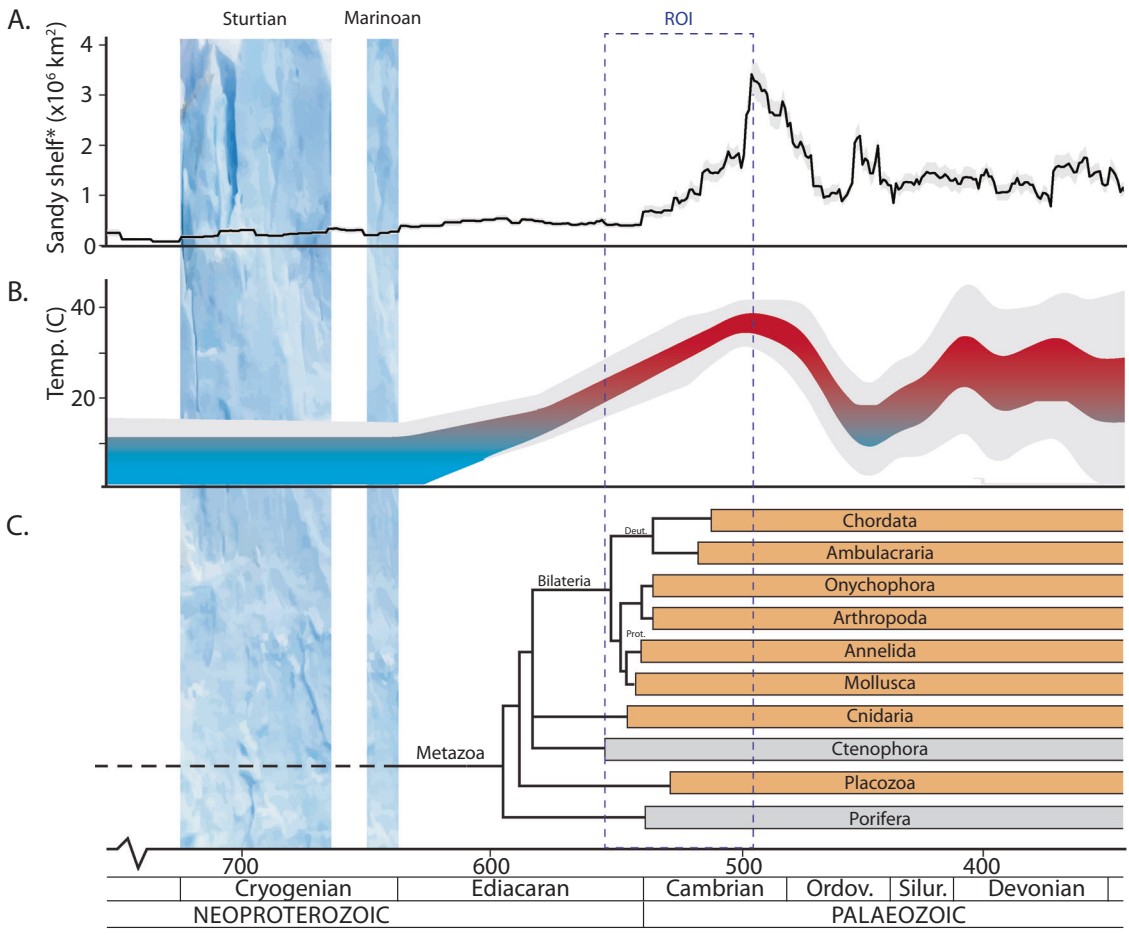

**Fig. 1 | Global temperature and animal diversification across the Neoproterozoic-Palaeozoic boundary.** At this boundary, the temporal extent of potentially global glaciations of the Cryogenian (blue vertical fields) is depicted as well as this study's region of interest (ROI). **A** The sandy shelf area in Laurentia and its increase in extent over the Neoproterozoic-Palaeozoic, from ref. 14 (black line, grey field represent bootstrap resampled error at ±1 std. dev.). **B** One modelled global average surface air temperature representation (at low latitude) modified from ref. 34, see also supplementary information and Supplementary Fig. 1 for comparison with clumped isotope data from ref. 27. Grey field represent ±1 std. dev

and +25 °C marking a transition from cold (blue) and warm (red) climate states. A gradual shift from an icehouse to a greenhouse climate (blue-red colour). **C** The divergence of animal clades (one of several possible time-calibrated trees, based on ref. 96, with approximate divergence between lineages based on ref. 97 and ref. 71). Boxes represent early branching clades and Bilateria that today perform cellular oxygen sensing mechanisms with the Hypoxia-Inducible Factor (HIF) system (filled orange) or not (grey), based on refs. 43,53. The positions of Ctenophora and Placozoa are based on ref. 96.

environmental fluctuations and resource availability above the needs of survival and maintenance[11]. For the Cambrian, however, it remains unknown, and we here aim to test, whether diel oxygen fluctuations would have been present to accelerate diversification[10].

The benthic Cambrian habitat in which animals primarily diversified[2] expanded with large-scale global flooding of continental fragments from Rodinia and Pannotia[12,13]. In Laurentia, sandy sediment deposits in shallow settings increased in areal extent four-fold during the early Cambrian Period, as demonstrated by macrostratigraphic summaries of rock quantities in North America[14] and consistent with strontium isotope data[12,15,16] (Fig. 1A). This flooding resulted in a dramatic expansion in sunlit shallow water sediments where benthic microalgae photosynthesis likely regulated the diel oxygen dynamics and seafloor redox conditions, as in modern seas[17,18]. Given that microalgae contribute substantially to the benthic oxygen budget in the modern coastal ocean[19,20] and became ecologically significant in the Cryogenian-Ediacaran[21,22], their presence can be presumed to have contributed to pronounced diel benthic oxygen fluctuations in the end-Neoproterozoic and early Palaeozoic. Notably, the microalgae photosynthetic oxygen production and oxygen consumption through heterotrophic respiration at the sediment-water interface is

fundamentally controlled by temperature[23]. Temperature also regulates diffusive transport mechanisms and oxygen solubility in seawater. With increasing temperature, diffusive transport mechanisms increase and oxygen solubility decrease that leads to faster transport rates but less availability of oxygen. During the Cambrian Period, global temperatures were generally defined by a greenhouse climate as indicated by modelling efforts[24,25], oxygen isotope palaeothermometry[26] and climatically sensitive lithologies[27–32] (Fig. 1B, S1). Annual equatorial sea surface temperatures have been estimated as high as 30–38 °C in the Cambrian[32]. Although reconstructions suggest a variable climate over the post-Marinoan[33], both shallow shelf area and global temperature increased between the Cryogenian and the Cambrian periods[14,27,34]. The higher temperatures would have increased photosynthesis-driven gross oxygen production, the molecular diffusion of oxygen, and the coupled oxygen consumption in benthic microalgae communities. Consequently, oxygen dynamics and variability in oxygen concentrations between day and night likely would have been amplified with higher temperatures, leading to temporally amplified daily redox fluctuations[35] and more physiologically stressful conditions for early animals in the expanding shallow benthic habitats.

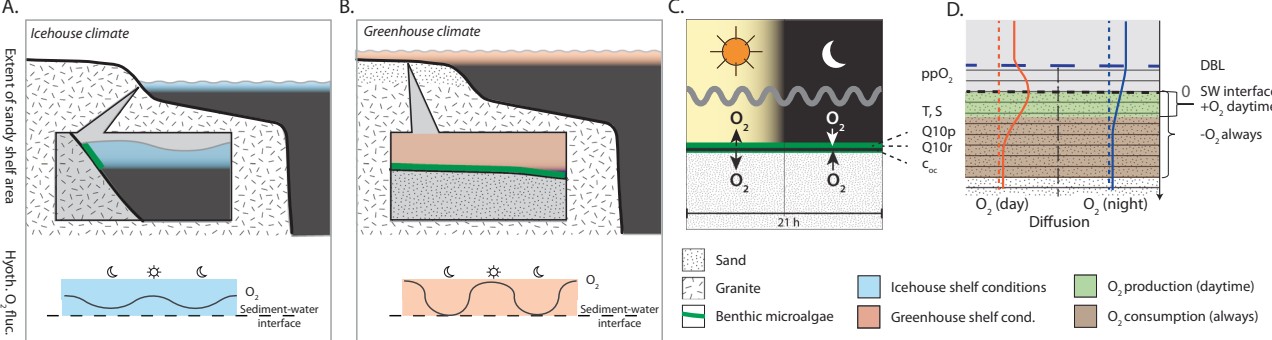

Fig. 2 | Projected changes in global shelf extent and local diel $O_2$ concentrations at the sediment-water interface. A schematic visualisation of the shallow sandy shelf and habitat expansion, the corresponding amplified extent of benthic microalgae habitats in the photic zone (green line), and the hypothetical diel $O_2$ dynamics at (A) the icehouse (blue) and (B) the greenhouse (beige) climate end-member scenarios, with cold versus warm shallow shelf settings. C A schematic understanding of benthic microalgae communities (green line) at the sediment-water-interface where it regulates sediment oxygen concentration dynamics by photosynthetic gross $O_2$ production during daylight and by heterotrophic $O_2$ consumption during nighttime. D The applied 1D biogeochemical model illustrating the diffusion boundary layer (DBL), the sediment-water (SW) interface and two modelled steady state $O_2$ concentration profiles as they appear during daylight (red line, photosynthesis-driven net oxygen productive) and during night (blue line; net $O_2$ consuming), respectively. Model parameters are atmospheric oxygen ($ppO_2$), temperature (T), salinity (S), production Q10 (Q10p), respiration Q10 (Q10r) and TOC adjustment factor ($c_{oc}$).

Physiological stress is a potential driver of the emergence of evolutionary innovations[10]. In addition to increasing genetic variation (e.g. mutations), environmental stress lets a population probe alternative phenotypic states[36]. Since beneficial stress-tolerant traits have high heritability under stressful conditions[11], increased variance channelled through existing functional systems can both account for similarity and lead to adaptive radiation[10]. For a scenario where environmental stress leads to adaptive radiation, however, it is suggested that fluctuations are predictable and combined with a surplus of energy to drive organisms' metabolism (beyond what is needed for survival and maintenance)[11]. In contrast, constant extreme physiological stress (e.g. saline pools) or low-energy environments (e.g. dark caves) would hinder phenotypic explorations and, therefore, adaptive radiation[10,36].

A specific trait that allows extant animals to tolerate environmental fluctuations is cellular oxygen sensing[37]. Oxygen-sensing mechanisms regulated by Hypoxia-Inducible Factor alpha (HIF-α) allow animals to orchestrate tissue homoeostasis by sensing and responding to redox fluctuations[37]. Oxygen shortage can be physiologically stressful for multicellular organisms by affecting metabolic homoeostasis[38], and extant animal cells coordinate a HIF-α response within minutes by, for example, shifting from aerobic to anaerobic cell metabolism[39]. This means that oxygen-sensing mechanisms accommodate stress-avoidance traits with high heritability under stressful conditions. Components in the current oxygen sensing mechanism have evolved from prokaryotes (e.g. the Per-ARNT-Sim or PAS domain)[40], to choanoflagellates (e.g. the prolyl hydroxylation domain or PHD)[41] and within animals[37].

Today, all animals with true tissues (excludes Porifera) and a benthic component to their life cycle (generally excludes Ctenophora) respond transcriptionally to $O_2$ fluctuations via HIF-α[39,42] (Fig. 1C). These animals share a joint core to this sensing by an oxygen dependent degradation domain (ODDD) of the bHLH-PAS protein where a proline can be targeted by a PHD protein (if $O_2$ is present) and hinder gene transcription (for details on Molecular dynamics within Hypoxia Inducible Factor (HIF), see supplementary discussion). However, the components of the pathway vary in their presence within animal groups and in terms of how they sense and respond to oxygen fluctuations. For example, an additional C-terminal transactivation domain (CTAD) can enable the protein to bind with transcriptional coactivators of and a factor inhibiting HIF (FIH) can limit the regulation of genes[43]. Differences in oxygen sensitivity and regulatory roles are also described for the two different prolines (P564 and P402) targeted by PHD, and an N-terminal transactivation domain (NTAD)[43]. These variations are likely to contribute to fine-tuning of both $O_2$ sensitivity and range of regulated genes[44,45], such that rudimentary oxygen sensing mechanisms could have been a functional but evolving system that channelled stress-induced adaptations.

A downside of oxygen sensing functionality via HIF-α is its energy cost for the cell. For example, the pathway involves a continuous production of a protein that is either degraded (in the presence of oxygen) or stabilised to induce a response (at oxygen shortage). Therefore, we here tested trade-offs for oxygen sensing as benefits and costs vary, against the suggestion that fluctuating benthic $O_2$ concentrations led to physiological stress in which improved HIF-α functionality optimised fitness.

In this work, we hypothesise that the combination of physiological stress and the substantial expansion of nutrient-rich sunlit benthic habitats from the late Neoproterozoic to the early Palaeozoic promoted phenotypic probing and, subsequently, the adaptive radiation of animals (Fig. 2A, B). We further hypothesise that stress-induced adaptations could have been channelled through the regulatory system of oxygen-sensing mechanisms. To explore this, we first investigate benthic $O_2$ fluctuations over day- and night cycles (Fig. 2C) at varying temperatures and carbon loads using a 1-dimensional (1D) biogeochemical model (Fig. 2D)[46]. We then assess trade-offs between the benefits of an efficient oxygen-sensing mechanism and its energetic costs for early animal fitness by modelling ecological (population) and evolutionary (strategy) dynamics of organisms with effective or poor oxygen-sensing mechanisms (eOSM or pOSM) existing in the same environment.

## Results

### Shifts in daily benthic $O_2$ fluctuations on an expanding shelf

Biogeochemical model: We applied a 1D biogeochemical model to constrain $O_2$ concentration dynamics and predict the concurrent physiological stress imposed by rapidly fluctuating (redox) conditions over a 21-h day and night cycle (early Palaeozoic)[47]. The partial differential equation model includes oxygen gross production, consumption and diffusive transport equations (Fig. 2C, Supplementary Table 1 for parameter values and supplementary methods for model description). The model results were validated against measured data from modern marine microphytobenthic studies at sunlit shallow-water sandy sediments (Supplementary Fig. 2). As the model is qualitatively generic and reflects key biophysical processes, identical dynamics and shape of profiles would be expected if validated against ocean shelf or

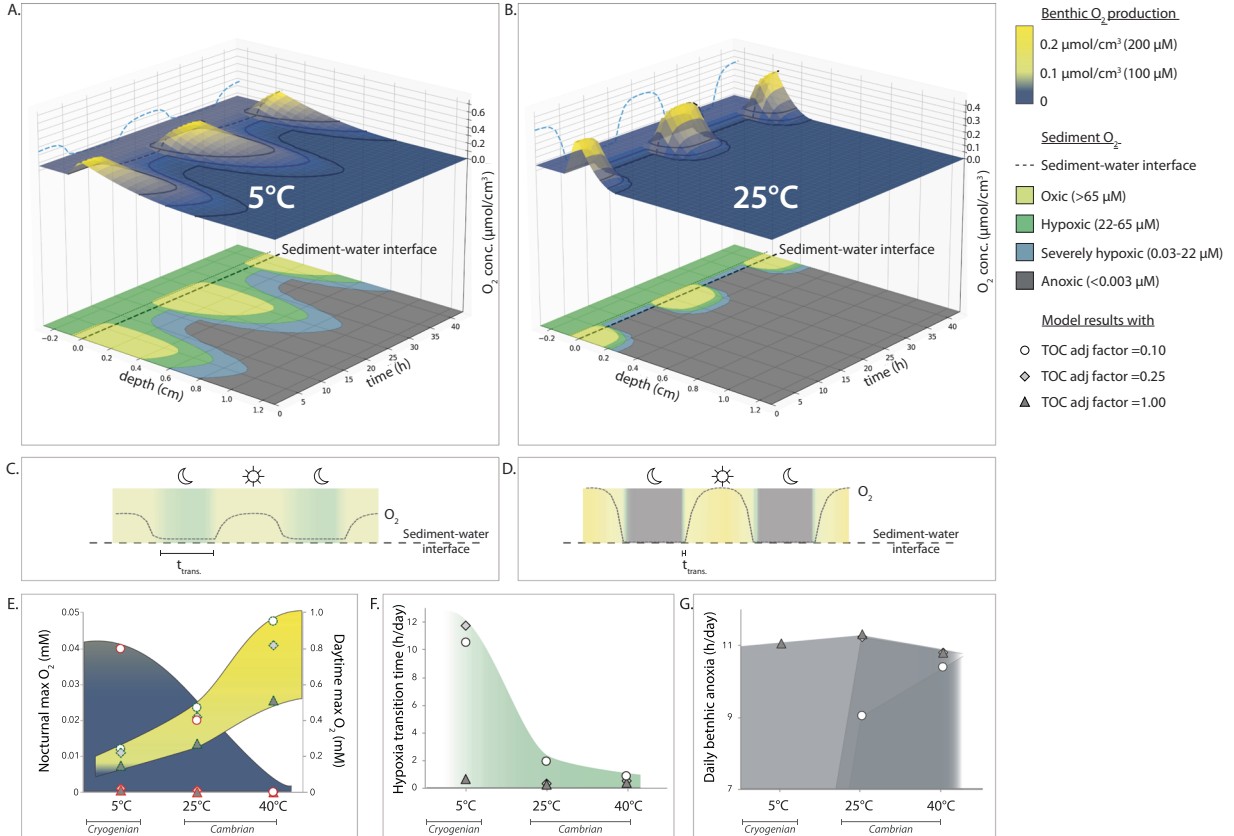

**Fig. 3 | Elevated local oxygen amplitudes within a sandy sediment as dictated by elevated temperature.** Diel O₂ fluctuations were modelled across the sediment boundary layer (0.05 cm) and 1.25 cm into the sediment at diel scales at (**A**) the cold scenario (+5 °C) and (**B**) the warm scenario (+25 °C). The illustrations include 2.5 days of 21 h each. Diel oxygen amplitudes during day (yellow) and night hours (green), and the transition time ($t_{trans}$; graded coloured) in hypoxic conditions that benthic animals would encounter become considerably more abrupt, going from (**C**) the cold to (**D**) the warm conditions. Also, TOC availability increased from the Cryogenian to the Cambrian with a factor of 0.1–0.8 relative to the Phanerozoic mean, respectively[3]. (**E-G**) Modelling results at +5 °C, +25 °C, and +40 °C and TOC settings of 0.1 (white circles), 0.25 (light grey diamonds) and 1 (dark grey triangles) relative to Phanerozoic mean are illustrated for (**E**) maximum benthic O₂ concentrations in the night and day, schematic field with colour codes for oxygen concentrations as in A-B; (**F**) transition times (h day⁻¹) through hypoxic conditions (green field); (**G**) extent of (h per day) benthic nocturnal anoxia (grey field). In (**E**), modelling results for maximum benthic O₂ concentrations at night (scale to the left) at the different TOC settings are marked with full red outline and emphasised with a blue field. Modelling results for maximum benthic O₂ concentrations in the day (scale to the right) at the different TOC settings are marked with dashed green outline and emphasised with a yellow field.

even deep-sea sediments[48]. The modelled dynamics are universal for muddy, mixed, carbonate and sandy sediments and include oxygen scavenging by, e.g. iron and sulphide reduction[35]. The model was used to derive day and night O₂ concentration profiles and to quantify the impact on these of changing temperature and total organic carbon (TOC) availability (see 'Method' section for modelled scenarios). Since rapid changes in O₂ concentrations can be physiologically stressful to benthic animals when adjustments to their ATP-turnover are less swift[49], we also quantified the transition time between day and night conditions.

Biogeochemical model results: These first-order experiments found that in a cold scenario (+5 °C), daily oxygen fluctuations and diel amplitude would have been modest with weak oxic conditions during the day and with hypoxic conditions during night lasting for ~11 h (Fig. 3A, C). In contrast, a warm model scenario (+25 °C)demonstrated profoundly amplified daily fluctuations with abrupt changes from fully oxygenated conditions during the day to true anoxia during the night in less than 0.3 h (Fig. 3B, D). Regardless of changes in TOC (0.1 to 1 relative to Phanerozoic levels[3]), a hot climate (at +25° C and +40° C, compared to a cold at +5° C) led to O₂ concentrations that were high during the day and low during night hours (Fig. 3E), shorter transition times with hypoxic conditions (Fig. 3F) and longer exposure to anoxia at the sediment water interface (Fig. 3G). A visual comparison of the results for 20 %PAL versus 50 %PAL at varying TOC adjustment factors (Supplementary Tables 3–4) depicts that a delicate balance of sub-modern $pp$O₂, sub-modern TOC and temperature could lead to severe diel redox fluctuations at the sediment-water interface (Supplementary Figs. 4–5). The robustness of the model results with respect to parameter values was tested by running a Monte-Carlo simulation (50.000 times over, see Methods for details).

## Evolutionary shifts to the sensing and responses to daily O₂ fluctuations

Population and phenotype model: After finding that daily benthic redox fluctuations on an expanding Cambrian shelf would have presented animals with severe daily O₂ fluctuations, we explored the ecological (population) and evolutionary (strategy) dynamics of species with effective and poor oxygen-sensing mechanisms (eOSM or pOSM), respectively, existing in these environments. To do this, we used a mathematical modelling framework named *G functions*[50,51] that simultaneously captures ecological (population) and evolutionary (oxygen-sensing strategy) dynamics over time. In other words, the same set of equations can qualitatively capture the interplay between changes in oxygen levels, ecological dynamics and species competition, and adaptation via oxygen sensing mechanisms. The capacity to effectively sense and respond to changing oxygen levels (rate of

phenotypic switching) presumably incurs a cost, but it would be an adaptive advantage for an organism to swiftly change from aerobic to anaerobic cell metabolism in this environment. For example, for anaerobic metabolism to produce ATP that matches that of aerobic metabolism (per glucose molecule), eighteen times more glucose is required. An increased metabolic rate is feasible, and the organism can remain highly functional, if also rate-limiting enzymes keep up. Expression of phosphofructokinase, the rate limiting and irreversible step in this pathway, is rapidly upregulated by HIF-α[49]. Organisms with eOSM would have the benefit of responding quickly to changing oxygen conditions, but at some cost. The costs of rapid cellular phenotypic plasticity could include cells refining oxygen sensing or maintaining the intra-cellular machinery to rapidly upregulate anaerobic metabolism (or their rate-limiting enzymes). Our model extends the classic Lotka-Volterra competition equations and imposes a cost for efficient oxygen-sensing. Namely, the population dynamics are given by Eq. (1):

$$\frac{dx_i}{dt} = x_i G(v, \mathbf{u}, \mathbf{x})|_{v=u_i} \qquad (1)$$

where the per capita growth rates of a species are given by Eq. (2):

$$G(v, u, x) = \frac{r}{K(v)}\left[K(v) - \sum_{j=1}^{n} x_j\right] - ds \qquad (2)$$

and the carrying capacity is a function of how well the focal individual's strategy matches the environment, Eq. (3):

$$K(v) = K_m \exp\left[-\frac{(v-\gamma)^2}{2\sigma_k^2}\right] \qquad (3)$$

In these equations, $v$ represents a species' cellular metabolism phenotype (strategy), $\mathbf{u} = (u_1, u_2)$ and $\mathbf{x} = (x_1, x_2)$ are the strategies and population sizes of each species in the population, respectively. The evolutionary dynamics are then derived by Fisher's fundamental theorem[52] as Eq. (4):

$$\frac{du_i}{dt} = s_i \frac{dG}{dv}|_{v=u_i} \qquad (4)$$

As a simplified L-V competition model, we assume that competition among species is independent of their strategies. All individuals have the same adverse impacts on each other and compete equally in a density-dependent fashion. This leads to growth in a logistic manner, wherein the carrying capacity depends on how well adapted the species is to its (continually changing) environment. Finally, we included a cost of oxygen sensing through the last term in the G-function. We assume this cost scales linearly with the capacity for oxygen sensing. The eOSM and pOSM species are differentiated solely by the eOSM species having a higher capacity for oxygen sensing, $s$, than the pOSM species. Thus, the eOSM species can switch its cellular metabolism more rapidly than the pOSM species. For further details on model derivation, please consult the 'Methods' section.

To capture the ecological (population) and phenotype (strategy) dynamics of each species (eOSM and pOSM), we performed three simulations, in which we let the optimal phenotypic strategy (γ) vary in a sinusoidal fashion with added stochastic noise. We simulated this over 500 time-steps (pseudo time; i.e. an arbitrary time unit) to capture the dynamics before the continental flooding, as in the early Palaeozoic. For simplicity and visualisation purposes, the flooding that led to the expanded shallow shelf area was implemented as a switch. To simulate the flooding, we increased the periodic $O_2$ fluctuations (driven by microalgal production and consumption), stochastic $O_2$ fluctuations (driven by nutrient cycling), or both.

Population and phenotype model results: The results of the simulations can be seen in Fig. 4A–C. When oxygen fluctuated stochastically, the benefit of tracking environmental oxygen levels with an efficient oxygen sensing was greatly outweighed by its cost and pOSM species dominated (Fig. 4A). When oxygen fluctuated periodically (as if daily), eOSM species dominated the population within ~500 (pseudo) time-steps after the switch, suggesting the benefit of perceiving and responding to redox fluctuations in a spatiotemporal manner (Fig. 4B). In the case of both periodic and stochastic fluctuations, eOSM species were greatly favoured, which drove the pOSM species to extinction within 250 (pseudo) time-steps (Fig. 4C). Although a qualitative estimate in pseudo-time, the biological event rate exceeds that of fluctuations, and both are instant in comparison to atmospheric changes of oxygen concentrations. The trends observed in the plots of eco-evolutionary dynamics generally hold for a wide range of parameter values (for details on Model of population dynamics and speed of phenotypic plasticity, see Supplementary discussion and Supplementary Figs. 7–8). In all cases, the pOSM and eOSM species attempt to track the environmental oxygen levels by adjusting their strategy. The eOSM species, which can more effectively track the oxygen levels due to their improved oxygen-sensing capability, displayed greater changes in strategy. Note that the cost of the oxygen sensing mechanism has a large influence on the outcome of the competition experiments (Supplementary Fig. 8). The competition experiments demonstrated that eOSM species are favoured across the range of costs of the mechanism tested in environments with stochastic *and* periodic fluctuations, and favoured for most tested costs ($d < 0.09$) in environments with periodic fluctuations. In contrast, pOSM species are favoured for most tested costs ($d > 0.01$) in environments with only stochastic fluctuations. The shape and magnitude of the cost function was chosen arbitrarily for the purposes of this qualitative modelling study, but for a given environment, there will be a threshold value for $d$, below which eOSM species will outcompete pOSM species and above which pOSM species will outcompete eOSM species. Furthermore, it is possible that extreme costs can overshadow the main trends observed in this in silico study.

The population and phenotype model is agnostic to which organisms would represent pOSM or eOSM species. To relate the phenotype model, the flooding event and early animal diversification in the Cambrian Period (Fig. 4D, E) to the early evolution of animals, we mapped differences in the composition of HIF-driven oxygen sensing across the animal clade based on sequences in ref. 53 and ref. 43 (Fig. 4F, supplementary discussion and Supplementary Table 4, for details). While HIFs that today compose the OSM of modern animals contain components (e.g. genes, domains, or proteins) that can be traced back to the origins of yeast and even to prokaryotes[37], its building blocks expand and differ in functions within and between early branching animals. For example, oxygen sensing in animals with e.g. only P564 would be simpler than in animals e.g. with P564 and P402 or with P564 and CTAD. Components to the OSM like the P402, CTAD, NTAD and Factor Inhibiting HIF (FIH) regulate a wide scope of target genes at a wide range of conditions, some of them with clear importance for development[37]. From what we know of the importance of these components in Bilateria[54–56], the different configurations amongst animals[43,53] reflect varying efficiency and costs to sense and respond to redox fluctuations.

## Discussion

By applying two lines of first-order reconstructions, we demonstrate that daily $O_2$ fluctuations in a photic benthic marine setting could be severe enough to induce physiological stress and promote specific adaptations among early animals at flooding events in the end-Neoproterozoic to early Palaeozoic. Biogeochemical modelling shows that the benthic biota on, for example the Cambrian shelf that expanded 4-8 times[14], could have been exposed to fully oxic conditions

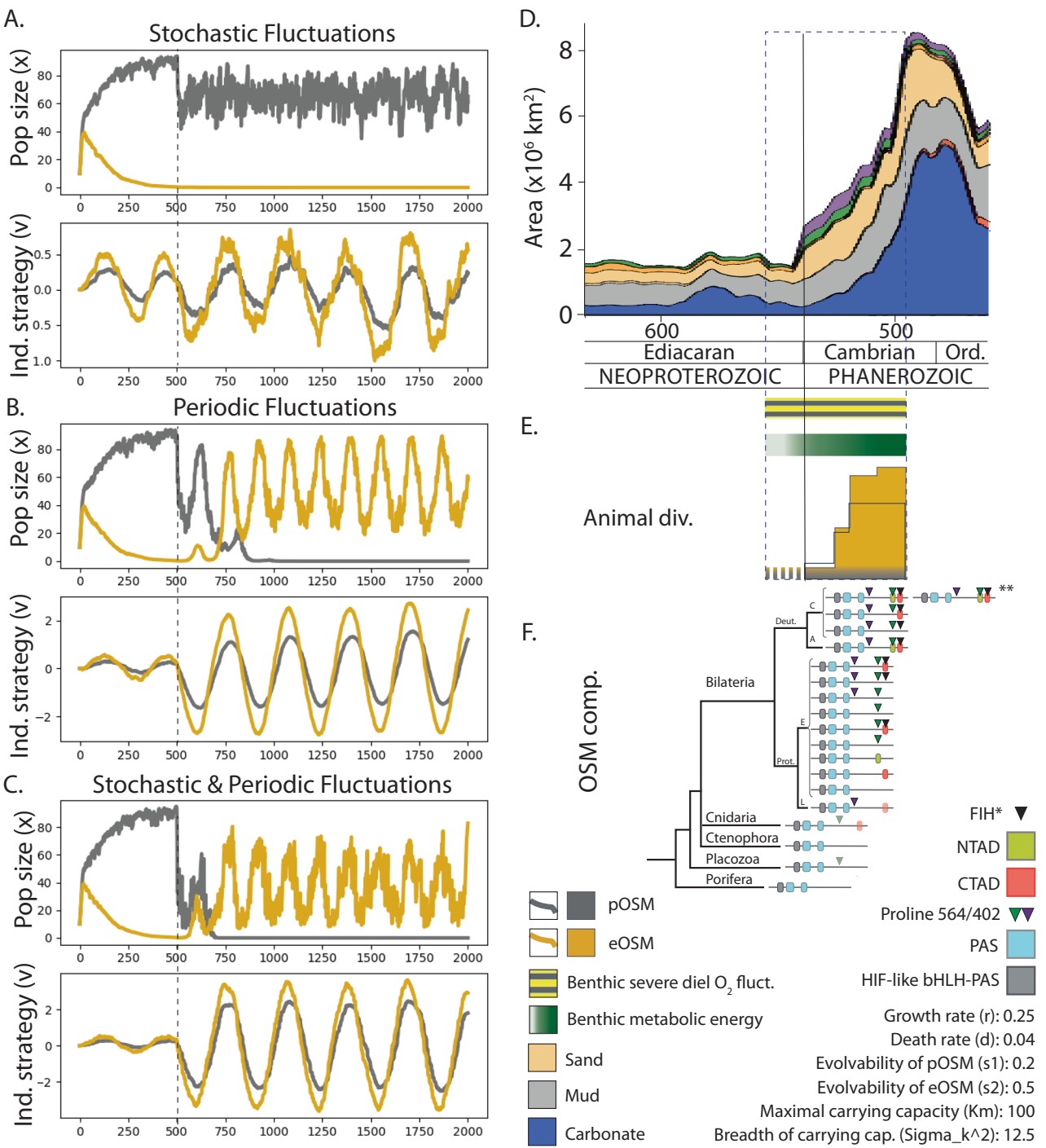

**Fig. 4 | Eco-evolutionary dynamics at the Neoproterozoic-early Palaeozoic boundary from a functional perspective.** Competitive eco-evolutionary dynamics between organisms with poor oxygen sensing (pOSM) and efficient oxygen sensing mechanisms (eOSM) demonstrate that an increase (dashed line) in the frequency of (**A**) stochastic oxygen fluctuations (driven by nutrient cycling) is not enough to favour the eOSM species, (**B**) periodic oxygen fluctuations (driven by microalgal production and consumption) eventually favour eOSM, and (**C**) of both stochastic and periodic $O_2$ fluctuations directly favours eOSM. **D** All sediment types deposited at the North American craton at the PC/C boundary as extracted from Macrostrat demonstrate an ~8-fold increase in areal extent of particularly shallow (sand and carbonate) settings (modified from[14]). **E** Schematic of our suggested cascade of events with stress (diel benthic redox fluctuations) developing with the greenhouse climate in the end-Neoproterozoic (striped black-yellow) and increase of metabolic energy (green) as a driver for diversification of animals (genus level) with a presumed distribution of species with pOSM (grey) and eOSM (orange). The schematic representation of animal diversity (number of genera, singletons

omitted) and disparity (number of classes, grey box) is modified from ref. 74. **F** Molecular differences in the HIF-like structures between the Porifera, Placozoa, Cnidaria, Bilateria and within Bilateria[43,53], with phylogeny of animal lineages based on ref. 96, with approximate divergence between lineages based[97] and ref. 71). A maximum diversity of components to the OSM is depicted per lineage in a simplified phylogeny[97] (to Lophotrochozoa (L), Ecdysozoa (E), Ambulacraria (A) and Chordata (C) for Bilateria), based on sequences in ref. 53 and ref. 43 (see *SI text* and Supplementary Table 4 for details). Proteins were not drawn in scale and the position of the components are approximate. The HIF-like bHLH (grey) and PAS (blue) is common to all, while the P564 (green triangle) appear from Cnidaria and Placozoa and CTAD (in Cnidaria, pale red when uncertain, see Supplementary Table 4) and Bilateria (red). Within Bilateria, multiple combinations of P564, CTAD plus P402 (purple triangle), NTAD (yellow, pale yellow when uncertain) and FIH (black triangle and grey when uncertain) are present. *asparagine 803 which is the FIH interaction site.** depicts human reference with also HIF-2α.

during the day to, after an abrupt shift, nocturnal anoxia (Fig. 3). In contrast, the modelling indicates that a cold setting would offer only modest daily $O_2$ fluctuations. More specifically, in a +5 °C setting (where atmospheric $O_2$ is set at <6% and TOC lower than on average in the Phanerozoic), the setting is never completely devoid of oxygen (weakly oxic in the day and hypoxic at night). Furthermore, our qualitative modelling of population and phenotypic dynamics demonstrated that investment in an energetically costly strategy (e.g. efficient oxygen-sensing or eOSM) pays off in fitness when species are exposed to periodic chemical fluctuations (Fig. 4B). The evolutionary success of animals with eOSM is even clearer when compounded with stochastic effects (Fig. 4C), such as those manifesting during algal blooms that could induce water column anoxia in the already poorly oxygenated Cambrian oceans[7,8]. Our results therefore call for a re-evaluation of factors driving the Cambrian explosion, highlighting how shifts in abiotic parameters and associated physiological stress over both spatial and short temporal scales could spur the adaptations observed in populations seen today.

## Stress-driven needs and adaptations

The significance of physiological stress across brief time spans in driving evolutionary innovation is frequently discussed, particularly in pivotal transitions throughout life's history[6]. For example, metabolic stress through both warm climate and oxygen variability is suggested for biotic turnover at the Permian/Triassic[57,58]. The Cambrian Explosion, however, has been treated differently, with studies focusing primarily on long-term changes in the atmosphere and the water-column (of e.g. $O_2$ or TOC) creating permissive conditions[3]. Our results pinpoint that seemingly inhospitable conditions—the fast and daily transitions to nocturnal anoxia—could act on the expanding Cambrian shelf setting. Since this respiration-driven nocturnal anoxia is primarily dictated by the temperature, it is worth noting that the paleotemperature record for this period is scarce. Evidence of global ice ages, however, anchors the coarse assumption that benthic $O_2$ dynamics changed from a colder mid-Neoproterozoic with limited shelf area to a warmer early Palaeozoic scenario with broadly flooded continental shelves. This means that even if flooding eroded evidence of larger Cryogenian shelves, benthic $O_2$ fluctuations in cold settings would have remained modest. In the Cambrian Period, flooded continents and a new benthic niche (produced by the new weathering regime of sands, muds and carbonate) at least to mid latitudes would have experienced the severe $O_2$ fluctuations between day and night. To inhabit the new fertile grounds, animals must have also tolerated associated physiological stress, paced by daily and severe oxygen fluctuations. To tolerate stress, animals manage metabolism.

Stress associated with recurring anoxia creates a need for animals to manage metabolism, like balancing the enzymes needed for glycolysis. Recurring anoxia also creates the need to manage other necessary functions. For example, anoxic nights associate with sulphide oozing from shelf sediments. Dissolved sulphides can diffuse across cell membranes without transporters (gasotransmitters) and are toxic to mitochondrial respiration[59]. Modern bivalves anchored in sulphidic sediments handle such exposure by an endosymbiotic relationship with sulphur-metabolising bacteria that convert the sulphide to less toxic forms[60]. Under these conditions, iron homoeostasis is also affected in multicellular eukaryotic organisms. To compare with oxic conditions and circumneutral pH, ferrous iron is oxidised and precipitates stepwise from nanoparticular Fe(III) crystallites to a crystalline ferric Fe(III) (oxyhydr) oxides[61]. Eukaryotic cells (like those of animals) transport this insoluble but bio-essential ferric iron across its membrane with mechanism like ferric-chelate reductase[62]. However, under anoxic marine conditions, reduced iron [ferrous or Fe(II)] dominates, meaning that animal cells risk exposure to iron deficiency[63]. Today, animals rely on the HIF system to mitigate these implications of exposure to anoxia. HIF-α participates in the upregulation of rate-limiting enzymes during glycolysis[49], sulphide tolerance[64,65] and fine-tuning of the cell's iron budget[62,63] and thus help secure a stable energy supply to cells. To compare, yeast regulate iron uptake, storage and use via three independently regulated routes and cannot mitigate iron fluctuations via a single mechanism[66]. Therefore, the adaptation of refined oxygen-sensing via HIF-α may have allowed animals to endure daily and stochastic benthic redox fluctuations.

Adaptations that early animals evolved to handle redox fluctuations remain unknown and hard to test. That the core of the HIF-α mechanism evolved once is clear[43], but also that several of its components to sense and respond to redox fluctuations appear and vary within the animal clade (for details on these components, see also supplementary discussion and Fig. 4F). For example, the ancestral components (e.g. PAS and bHLH domain proteins)[40,41,67] with the addition of a proline targeted for hydroxylation (P564) are described as present in Placozoa and Cnidaria, whereas the CTAD is undoubtedly present first in Bilateria[42,43] (see Fig. 4F and Supplementary Table 4 for comparisons).

In bilaterian animals, however, the variation of combinations increase with e.g. a second proline (P402) that can be hydroxylated and genes transactivated via the N-terminal or NTAD[45,68,69]. Within the NTAD, the interaction site for the Factor Inhibiting HIF (FIH) is also noted within Bilateria. FIH can hydroxylate an asparagine residue at $O_2$ tensions that are higher than when the proline (P564) is hydroxylated[70]. This means that animals with FIH can demonstrate HIF-α that is stabilised or hydroxylated (and therefore degraded) differently than in animals without FIH. Based on how FIH, CTAD, NTAD and the prolines are known to differentially register and respond to $O_2$ fluctuations[54–56], cellular oxygen-sensing mechanisms appears be become refined between Placozoa and Cnidaria, between Cnidaria and Bilateria and especially within Bilateria (compare e.g. variants within Protostomia in Fig. 4F). The known differences and functions of these components could have been separating Placozoa (pOSM) from Cnidaria (eOSM), or species with pOSM from eOSM within cnidarians or bilaterians; whether living in the Ediacaran[71] or Cambrian. If so, the environmental stress in the Ediacaran and early Palaeozoic can be connected to a physiological impact and to adaptations within and between animal groups that led to diversification through novel niche filling. As discussed above, to achieve rapid cellular phenotypic plasticity would come with a cost and eOSM species can in specifically periodic conditions bear that cost. This leads us to see that the evolution of traits like oxygen sensing and oxygen responses may have important functional roles that could have contributed to animal diversification. Therefore, our population and phenotype model results combined with previous work on the differences within the molecular differences in animal OSM are consistent with the view that adaptations of rudimentary oxygen-sensing capacities gave certain animals a competitive advantage over other early metazoans to cope (by phenotypic switching) within the opened shallow sunlit niches with diurnal benthic oxygen fluctuations.

## From physical harshness to diversification

Animal diversity today follows latitudinal gradients, largely attributed to the tropics as being more productive[72]. The higher productivity offers extensive habitats and biotic feedbacks where species themselves create environmental heterogeneity that, in turn, promotes the evolution and coexistence of even more species[72]. When an ecosystem is highly productive and incurs high biotic stress through competition, predation and competition between different mutualistic associations, a kind of bio-diversity pump ensues. Therefore, a setting with high productivity like the tropics today promote and can tolerate tighter niche packing that involves rapid and efficient use of available nutrients and resources from the lower to higher trophic levels. However, tight niche packing of an ecosystem does not necessarily promote diversification. When an ecosystems instead is governed by abiotic and

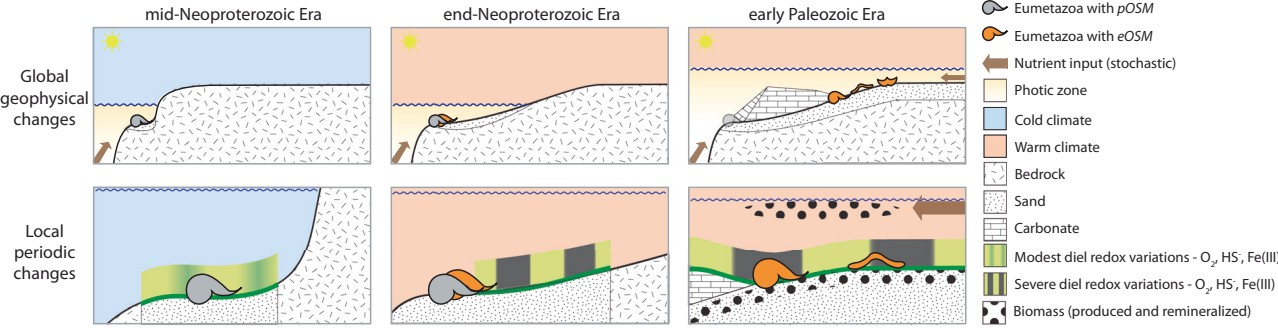

**Fig. 5 | Global and local geobiological changes converge to drive the diversification of primarily HIF phyla.** In the mid-Neoproterozoic, a narrow and unchanged shelf habitat offered icehouse-dictated (blue for cold temperatures in air and water) modest diel benthic fluctuations (left panels). Towards the end-Neoproterozoic, warmer climate (pink for warm temperatures in air and water) increased amplified diel redox fluctuations and physiological stress while the unchanged habitat and low productivity maintained harsh ecological conditions (middle panels). In the early Palaeozoic, global flooding of shallow shelf area with the severe diel benthic redox periodicity expands dramatically, thus lifting the biological stress by surplus of metabolic energy in the short term (right panels). The combination of stress-induced needs for oxygen-sensing and increased metabolic energy led to new needs, adaptations and radiation of animals with adequate oxygen sensing mechanism, e.g. sponges (without HIF-α) and Bilateria (with HIF-α).

physical stress, adaptations strive to mitigate physical harshness at the expense of traits aimed at suppressing competitors and avoiding predation[11]. These new needs and adaptations are independent of population density. Indeed, the density-independent needs of species under physical harshness generate less intense frequency-dependent interactions that can promote species diversification[73,74], especially when combined with a surplus of metabolic energy[11].

The shelf habitat of the say Cryogenian Period would have offered a fairly constant area and metabolic energy for as long as there was an icehouse climate[27,34] (Supplementary Fig. 1), little need for oxygen sensing adaptations (modest fluctuations). Metabolic rates, metabolic scope, and the need to rapidly pre-empt others for resources would have been low. The overall ecological environment would have been harsh for early animal life even if the chemical environment was not. In fact, the modest fluctuations in oxygen would have been the result of low primary productivity and turnover of organic matter. With the warm climate and enlarged continental shelves of the Ediacaran (as a shift from possibly one[75,76] or several[77,78] extensive glaciations) and Cambrian periods, primary productivity would have increased dramatically both in terms of per unit area as well as more area exhibiting high productivity. However, a consequence of this primary productivity would have been biotically driven stress from severe daily $O_2$ fluctuations – an additional physical stressor for animal life and a drag on diversification. Hence, we see the evolution of efficient handling of oxygen fluctuations (as via HIF-α) as constrain-breaking adaptations that mitigated the stress of benthic oxygen fluctuations, permitted fuller utilisation of the primary productivity, shifted the system from one of primarily physical harshness to one of extreme biotic harshness, and unleashed the biodiversity pump of the Cambrian Explosion.

### Diversification with and without HIF-α
Animals without HIF-α (e.g. sponges[79]) and with HIF-α (Placozoa, Cnidaria and Bilateria) diversify in the Cambrian[43]. Cnidaria are represented as early as in the Ediacaran biota[80], thus implying that animals with a core to the HIF-α pathway (e.g. P564 in the ODDD targeted by PHD) were present from before continental flooding in the Cambrian Period. If the HIF-α pathway indeed mitigated stress for early animals as it does in extant Bilateria (like upregulating rate-limiting enzymes for glycolysis)[49], it is fair to assume that heritability to modification of the HIF-pathway would be high. HIF target genes, furthermore, regulate cell differentiation in Bilateria, which is key to tissue architecture and homoeostasis[81,82]. Indeed, cell differentiation is one out of four key characteristics to underpin most animal phenotypes that have been realised over the Phanerozoic[83]. This would indicate that HIF, at least in

Bilateria, allowed for metabolic versatility and adaptations that also pertain to body plans (i.e. disparity). In contrast, sponges diversified without a HIF-α pathway and constitute a negative control to the hypothesis that HIF-α was key to the Cambrian explosion. While being diverse, sponges appear to lack the ability to form true tissues, as their cells generally do not connect in basement membranes[84,85]. Instead, the limited number of sponge cell types[86] demonstrate high plasticity through trans-differentiation[84,85] and mobility from outer to inner (mesohyl) layers[87]. Sponges also appear to rhythmically open and close towards the environment, which regulates internal $O_2$ from oxic to anoxic[88,89]. The interval of opening and closing the osculum differs between species (e.g. of ~40 min or 80–170 min intervals), which leads to canals being re-modelled each time the sponge contracts and relaxes again[88,89]. Although it remains unclear if the rhythmic changes to internal $O_2$ also alter the fate of sponge cells, it is clear that cell differentiation is an oxygen-driven process[90]. Therefore, one can speculate that pulses of internal anoxia (for stemness) and oxic conditions (for cell differentiation) represent a unique adaptation for regulating cell fate and maintenance in sponges. With an adaptation to control fluctuations internally, sponges would have bypassed the need for cellular OSM and diversified in their own right as density-independent needs (from abiotic stress) and access to metabolic energy changed with the Neoproterozoic to early Palaeozoic flooding events.

### The interaction of abiotic, ecological and functional factors for the Cambrian explosion
Based on our results, we offer the general principle that global and long-term geophysical changes converged with local and short-term biotic periodicity in a habitat with increased metabolic energy to drive the Cambrian explosion of primarily phyla possessing HIF-α (Fig. 5). When eustatic sea-level was high under the greenhouse climate, shallow sands and new carbonate platforms offered animals a rich marine 'tableland' but with severe abiotic harshness. At points, these conditions could have become too harsh, possibly explaining Cambrian intervals that are barren of animal fossils[91]. That physiological stress was involved in the Cambrian diversification may also be indicated by how extinctions were proportionally higher[92] than later in the Phanerozoic and generic longevity much expanded over the earliest Palaeozoic (from 0.8 myr in the Cambrian to 6.3 myr in the Ordovician)[93]. This data may be taken to reflect how a stressful environment was intertwined with both death and new adaptations. However, the overall success of animals with efficient handling of the redox fluctuations would have reached beyond simply enduring this setting,

allowing animals with specific adaptations, like sponges or Hifozoa[43], to diversify.

Previous explorations into the animal diversification in the Neoproterozoic-early Palaeozoic have largely overlooked the combined role of daily benthic redox fluctuations and physiological stress with changed metabolic energy as drivers. Recent observations, such as how early animals diversified *despite* globally low or oscillating $O_2$[5,33], inspire a shifted perspective on stress-induced evolutionary change.

In summary, our implementation of a biogeochemical model, paired with geological data, led to predictions that are consistent with the fossil record. Our results suggest that severe daily redox variations in an expanding sunlit benthic shallow shelf created niches in the end-Neoproterozoic and early Palaeozoic that could have promoted an adaptive radiation of organisms with efficient handling of oxygen fluctuations, such as those regulated by HIF-α. Our G function model that captures environmental oxygen fluctuations, population dynamics and species competition, and adaptation via oxygen sensing mechanisms demonstrated how periodic fluctuations could favour species with efficient oxygen sensing mechanisms, despite its metabolic costs. Further studies may detail low-oxygen environments that prevailed in the shallow shelf and their adjoining biotic adaptations, emphasising the role of daily benthic redox stress in shaping the animal evolution in the benthic environment.

## Methods

Biogeochemical model: Model parameters were tuned for sandy sediments and to mimic cold (5 °C) and warm (25 °C, and 40 °C) shelf conditions with 20–50% of present atmospheric levels (%PAL) of $O_2$ and 0.1–1 times the Phanerozoic load of organic carbon. While the term hypoxia was originally used to describe internal stress on an animal, it has also come to describe the external ocean medium[94]. The specific definitions of anoxic to oxic conditions used here are: anoxic <0.02 μM, severely hypoxic between 0.02 and 22 μM, hypoxic between 22 and 65 μM and oxic >65 μM[95].

To test whether the model results were robust with respect to parameter values, we ran a Monte-Carlo simulation of the model 50,000 times over a large range of $ppO_2$, TOC and temperature responses for production and respiration rates ($Q_{10}$Prod and $Q_{10}$Resp), accommodating for most possible seafloor conditions (Supplementary Table 1). These simulations supported the generality of the model by demonstrating that a short transition time (<1 h) from daytime oxic to nighttime anoxia/severe hypoxic conditions in the diffusive boundary layer (DBL) was met in 67% of the cases and that the model output is primarily sensitive to TOC adjustment factors and $ppO_2$ (various scenarios shown in Supplementary Fig. S6). A compilation of TOC contents in more than 13,650 shale samples from the end-Neoproterozoic and Phanerozoic eons indicates that TOC in the Cryogenian were lower than in the Cambrian Period and the Phanerozoic overall[3]. Quantitative differences arise if parameters are set for muddy, mixed, or carbonate sediments, but the same fluctuating patterns and qualitative trends remain alike[48].

Population and phenotype model: The G function framework we use allows us to track both how the population sizes of the species change over time (ecology) as well as how their strategies evolve in response to changing environments (evolution). We modelled the population dynamics of two species, one with high phenotypic plasticity for cellular metabolism in response to dissolved oxygen conditions and one with low ($i = 1,2$), Eq. (1):

$$\frac{dx_i}{dt} = x_i G(v, \mathbf{u}, \mathbf{x})|_{v = u_i}$$

where the fitness-generating function, $G(v, \mathbf{u}, \mathbf{x})$, describes the expected per capita growth of a species as influenced by its cellular

metabolism phenotype (strategy), $v$, the strategies of each species in the population, $\mathbf{u} = (u_1, u_2)$, and the population sizes of each species, $\mathbf{x} = (x_1, x_2)$. In this case, the relevant strategy describes an abstract trait that allows cells of the multicellular organism to switch along a continuum between anaerobic and aerobic metabolism. Thus, this equation captures the change in the population size of a given species as a product of its current population size and its per capita growth rate. We let the rate of phenotypic switching of cells of the organism depend on (a) the slope of the fitness gradient and (b) how fast the species can change phenotype in response to the fitness gradient. The slope of the fitness gradient is given by $\frac{dG}{dv}$. How fast the species can change in response to this fitness gradient depends on the oxygen sensing mechanism; the more developed this is, the more quickly a species can switch. These components together allow for the derivation of the equation for phenotype dynamics[52], Eq. 2:

$$\frac{du_i}{dt} = s_i \frac{dG}{dv}|_{v = u_i}$$

where $s_i$ is a measure of the capacity for cellular oxygen sensing that allows the organisms to closely track and respond to their environment. To define our fitness-generating function, $G$, we used simplified Lotka-Volterra (L-V) competition equations, Eq. 3:

$$G(v, u, x) = \frac{r}{K(v)}\left[K(v) - \sum_{j=1}^{n} x_j\right] - ds$$

We let the carrying capacity be a function of the focal individual's strategy. We assume that there is some optimal strategy value γ. The closer the individual's strategy is to γ, the higher its carrying capacity, Eq. 4:

$$K(v) = K_m exp\left[-\frac{(v - \gamma)^2}{2\sigma_k^2}\right]$$

We assume that deviations of strategy values from γ decrease carrying capacity in a Gaussian fashion. We allow γ to change over time as a result of fluctuating oxygen levels in the environment. A particular dissolved oxygen level corresponds monotonically to a particular value for γ. In the model, the ecological impact of a species' phenotype to oxygen levels comes through its carrying capacity. As a simplified L-V competition model, we assume that competition among species is independent of their strategies. All individuals have the same adverse impacts on each other and compete equally in a density-dependent fashion. This leads to growth in a logistic manner, wherein the carrying capacity depends on how well adapted the species is to its (continually changing) environment. Finally, we included a cost of oxygen sensing through the last term in the G-function. We assume this cost scales linearly with the capacity for oxygen sensing. The eOSM and pOSM species are differentiated solely by the eOSM species having a higher capacity for oxygen sensing, $s$, than the pOSM species. Thus, the eOSM species can switch its cellular metabolism more rapidly than the pOSM species.

### Reporting summary

Further information on research design is available in the Nature Portfolio Reporting Summary linked to this article.

## Data availability

Supplementary information and code are provided. The biogeochemical model is implemented using the Mobius framework[46], and the source code and files are openly accessible. The eco-evolutionary model is implemented using the G function framework[50,51], and the source code is openly accessible.

## Code availability

The biogeochemical model is implemented using the Mobius framework, and the source code can be found in Supplementary Code 1 and at: https://github.com/NIVANorge/Mobius/tree/master/Modules/SedimentOxygen. Files used for running the model can be found at: https://github.com/NIVANorge/Mobius/tree/master/Applications/SedimentOxygen. The source code to the eco-evolutionary model can be found at in the Supplementary Code 2.

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

## Acknowledgements

We thank Robert R. Gaines, Christopher Scotese, James Sample and Robert Austin for continuously helpful input and criticism. We are also grateful to members at the Tissue Development and Evolution (TiDE) group, the Department of Geology at Lund University and the Nordic Center for Earth Evolution (NordCEE) at the University of Southern Denmark for help with both conceptual and practical experiments. We thank the Norwegian Institute for Water Research (NIVA) for the access to the data platform and the Mobius framework. The authors are grateful for funding from the European Research Council (ERC) under the European Union's Horizon 2020 research and innovation programme, grant agreement No 949538 (E.U.H., M.I., E.B.); Norwegian Institute for Water Research (K.H.); VILLUM FONDEN grant 15397 (N.R.P.); Crafoord Foundation Grant 20220633 (A.B., C.C., E.U.H); The Royal Swedish Academy of Sciences MG2022-0019 (A.B.); The National Science Foundation Graduate Research Fellowship Pr. 1746051 (A.B.).

## Author contributions

Design of the study: E.U.H., K.H.; Building and calibration of the 1D diffusion model: E.U.H., K.H., M.D.N.; Building the G-function framework: A.B. J.S.B.; Performing experiments/modelling: E.U.H., K.H., M.D.N.; A.B. M.I.; Composing the HIF component map: M.I.; Analyses of results: E.U.H., K.H., M.D.N.; M.I., S.E.P., N.R.P., A.B., K.J.P., S.R.A., C.C., M.I., R.A.G., E.B. J.S.B.; Writing the manuscript: E.U.H., A.B., K.H., M.D.N.; N.R.P., S.E.P., C.C., E.B. K.J.P., S.R.A., R.A.G., J.S.B.

## Funding

## Competing interests

The authors declare no competing interests.
