## [Transparent Peer Review file · Nature Communications]

Benthic diel oxygen variability and stress as potential drivers for animal diversification in the Neoproterozoic-Palaeozoic

Corresponding Author: Dr Emma Hammarlund

Version 1:

Reviewer comments:

Reviewer #3

(Remarks to the Author)

I am honestly a little disappointed to be reviewing this manuscript for the fourth time and finding the authors disregard my suggestions yet again. I really am just trying to help the authors anchor an interesting modelling approach in a representative and up-to-date reading of the Ediacaran-Cambrian literature. I have outlined many suggestions about how the Ediacaran biota relate to eumetazoan clades in previous reviews, so I won't repeat them here. I suggest a reframe of the discussion to avoid making poorly supported overstatements about a driver of the Cambrian explosion and instead describe an interesting mechanism that may have impacted animal diversification at some point between the Cryogenian and Cambrian.

As per my previous reviews, I remain critical of the authors' overly confident descriptions of Cambrian vs Cryogenian climate and evolutionary trajectories. All that I think is really required here is a broader acknowledgment and appreciation that these mechanisms may have been important at some point between the Cryogenian and Cambrian but that we are not currently able to reconcile exactly when. I think that uncertainty is fine in the context of an interesting modelling study and don't really understand why the authors continue to try to overinterpret their results in an outdated evolutionary and climatic framework here.

I agree with the similar comments of reviewer 1 and really struggle to understand why we have gone through all of these rounds of review when the authors simply could have taken our suggestions on board. I can't help but feel that it is a waste of everyone's time to be debating this in rebuttal letters when the authors could have incorporated our referenced suggestions with ease. New analyses may be reasonably out of scope, but a discussion reframe would have been straightforward for the authors to incorporate at any juncture.

In its current form, I cannot recommend this paper for publication in Nature Communications. As emphasised repeatedly in my multiple reviews, fixing these fundamental issues really wouldn't be that hard but (unfortunately) the authors don't seem willing to do so.

Line 37 – I don't see the relevance of the reference to cancer here and strongly suggest removing.

Line 61 – interpretation of Sr isotope records is contentious. They certainly don't "demonstrate" this. I suggest you qualify the statement or remove.

Line 86 – "coupled to" – correlated with? A potential driver of?

Line 130 – specifically in the Cambrian? Why not anchor your hypotheses on this broad distinction between the Cryogenian and early Paleozoic. The more specific your time references here, the less compelling your narrative, in my opinion.

Line 326 – "we conjecture that oxygen sensing mechanisms evolved within early animals at the Cambrian flooding event" – I don't understand how you can invoke oxygen sensing evolution in early animals at ~520Ma (critical objection being the use of 'early'). Yet again, I feel like the discussion here is completely missing an informed perspective on the divergence times of

early animals. Surely this is way too late for these higher level divergences to be associated with innovation in OSMs? If you disagree then please justify.

Figure 4 – as I have said in every review of this paper, I firmly feel that the evolutionary timelines invoked for different metazoan lineages here is very out of date. For example, when we look at the Bilateria illustration in Fig 4F, what about the Ediacaran bilaterians? Dickinsonia, Spriggina and bilaterian trace makers are all fully disregarded here so far as I can tell. Similar stories can be applied to your Cnidaria and Porifera extents.

Lines 482 onwards – refer to above and previous comments on how out of date I feel this discussion of different lineages is. I strongly recommend that this would have to be updated to represent current understanding of the history of these lineages in order for this manuscript to be published in this form. I have provided detailed arguments on this in previous reviews and really don't understand why the authors are so resistant to doing this. If they fundamentally disagree with the literature that I cite then it would be useful to justify this in the text and put forward an argument that is neutral to these reconstructions. Currently this out of date discussion just weakens a perfectly good modelling study in my opinion, and will only frustrate readers in the Neoproterozoic community. Having spent the time to outline these arguments to the authors multiple times between my four reviews of this manuscript, I honestly am quite frustrated that they have not taken the small amount of time it would take to incorporate this more up to date discussion into their narrative.

Line 511 – “For the first time, a cascade of physical, functional, and ecological factors aligns directly with the Cambrian Period.” I would argue this is patently untrue. Why overreach like this? There is a cool underexplored mechanism here that you have the opportunity to tell the reader about. Why frustrate the reader by making claims like this that are clearly dismissed by a quite superficial understanding of the literature?

Fig 5 caption – see notes above.

Reviewer #4

(Remarks to the Author)

The authors have done a reasonable job of addressing the substantive points from my previous reviews. I would be happy to see this manuscript published following a few minor revisions which I still think are necessary and which I hope the authors find helpful.

There are some line-by-line comments below, but first I would strongly encourage the authors to fully revise the text to separate their general, first order, modelling results from their interpretation of how this would play out in the Neoproterozoic-Phanerozoic transition. In particular, this is very generally applicable, first order modelling. At various places, the text could be clarified to show that:

a. The first order modelling of oxygen concentration cycles was done for a range of T, TOC, and [O₂] conditions which could be interpreted broadly in a geographical and temporal way (e.g. for various places on present-day Earth or times in the geological past). Keeping this distinction clear throughout would help emphasise the broader applicability of the concepts modelled here and would widen the appeal of the study.

b. The first order modelling of pOSM and eOSM species could also be shown as of much broader relevance than it is currently presented as. In particular, the authors acknowledge that there are different differences in OSM efficiency between different animal clades (bilaterians, cnidarians, placozoans, poriferans, ctenophores). The focus in the results section on presenting this work as predominantly about the differences between bilaterians and other animals hides the wider implications which could be seen more broadly in modern and other palaeo-ecological settings. To keep to the early animal example, such an approach could be applied to the redox oscillations (on a different time scale, but same principle) of the later Ediacaran world, rather than just the Cambrian.

Finally, I do hope to see this published soon and I think the authors are very nearly there with it.

Line-by-line comments:

Line 9: Proterozoic-Phanerozoic, or Neoproterozoic-Palaeozoic, would be preferable to Precambrian-Cambrian.

Lines 27 to 28: ditto Line 9.

Fig. 1: Please define the dashed box label “ROI”.

Line 51 (Fig. 1 caption): rephrase to make one sentence about the blue-red transition; it doesn't currently flow. E.g. “+25 °C marking a transition from icehouse to greenhouse climate states”. Otherwise, the caption is much improved, thanks.

Lines 72 to 74: climatically sensitive lithologies have also contributed substantially to this picture (e.g. Boucot, A.J., Xu, C., Scotese, C.R. and Morley, R.J. 2013. Phanerozoic Paleoclimate: An Atlas of Lithologic Indicators of Climate, 1st ed. SEPM Concepts in Sedimentology and Paleontology 11).

Lines 74 to 76: something has gone wrong here, I think. Please check for grammar and punctuation.

Line 109: “ctenophores” or “Ctenophora”.

Line 117: this paragraph is good but, as someone who is not a biologist or biochemist, I only follow the implications of it through having already reviewed the paper. At Line 117 you could be a bit more explicit that each of the additions to the HIFa pathway increased its efficacy and so represent step-wise acquisitions of O₂ sensitivity across the animal tree.

Fig. 2: I reiterate that the maps in A and B should have the same colour scales – they are representing different temperatures of water mass, but still water mass. At present, this looks more like topography than flooded continental area and misrepresents that the continents were not completely flooded in the Cambrian. I still think the maps add confusion as this is not a latitude, or even geography, based study, but more about flooded continental shelf. The onshore-offshore profiles show this better. This is particularly pertinent as the next section talks immediately about a 1D model, no horizontal (map) resolution needed.

Line 168: “or anoxia” is superfluous. Just need “O₂ concentration.” As levels oxygenation are defined below (lines 172 to 174).

Lines 168 to 169: I don't follow the grammar of “Since quick O₂ changes require benthic animals to conform their ATP-turnover”. Should this be “conform to their ATP-turnover rate”? or something else?

Lines 176 to 178: I thought from their last response that the authors were refraining from calling the scenarios “Cryogenian” and “Cambrian” and instead sticking to “cold/warm/hot” or “icehouse/greenhouse” and then interpreting these later as reflecting Cryogenian and Cambrian. Presenting it in this way would emphasise that this is a first-order simulation experiment, not an actualistic one, particularly as some Cryogenian intervals were quite possibly greenhouse or super-greenhouse climate states. This would also emphasise the broader applicability of the work: this is just looking at pO₂ variation under different T, TOC, [O₂] conditions – these apply in the modern world (e.g. by ocean basin, by latitude) and in numerous other deep-time contexts. The generality of this is a strength that could be much better shown.

Fig. 3: I again urge the authors to make this figure easier to follow and assess. It should be split up or made bigger, or something. I also suggest again plotting the blue and yellow bands in Fig 3E either on the same scale or on separate plots: their numeric range is two orders of magnitude different and it is still not obvious without the caption which coloured envelope relates to which y-axis scale, particularly as the same blue and yellow are used for different oxygen concentrations in figs 3A & 3B.

Line 319: “d >= 0.01” includes “d >= 0.09”. I think a “, respectively” is needed to clarify that these values relate to stochastic and periodic fluctuations, respectively?

Line 332: “cnidarians” (lowercase “c”) or “Cnidaria”.

Lines 326 to 342: here is an opportunity to illustrate the impact of stepwise acquisition of increasingly ‘e’ OSMs, with each clade being slightly more ‘e’ than those earlier branching clades. I think I made a similar point before, but here the key is not the ‘bilaterians are best’ but that each clade made some slight improvements in their OSM. The text of this paragraph could again be strengthened by saving until the end of the paragraph the specific case of early animal evolution through the Neoproterozoic-Phanerozoic transition and keeping the initial interpretation as hypothetical discussion of organisms with less (‘p’) or more (‘e’) efficient oxygen sensing mechanisms.

Fig. 4: This is a lot better, and I really like the addition of panel F. I do again ask about the colour variation in the diel bar – are we to read anything into the yellow-grey-green transitions and their placement?

Fig. S1: this is really hard to interpret, particularly for anyone who has difficulty with their colour vision. I would urge the authors again to simplify this and use colour palettes that are robust to common colour vision deficiencies.

Version 2:

Reviewer comments:

Reviewer #3

(Remarks to the Author)

Firstly, apologies to the authors for the delay in submitting my review at the end of a busy year. I am genuinely really pleased to see the changes that the authors have made in response to my comments and am glad that some directness seems to have paid off in this case. I think the increased generality and nuance here will (despite making broader claims) lead to a much more well received manuscript when the community have the opportunity to read it. Now that the authors have addressed so many of my key (hopefully constructive!) criticisms, I am happy to recommend this manuscript for publication in Nature Communications without any further rounds of review.

Reviewer #4

(Remarks to the Author)

Firstly, the text is much clearer this time round, and I was pleased to read an improved version. Secondly, I don't have a problem with the fundamental science of this work (though I remain perplexed about why the results of the 40 C simulations are not really included), and I think it does have the broad relevance suitable for Nature Communications. Thirdly, however, I do still feel that there are improvements needed regarding how the work is contextualised and presented, but I think most of these are more personal preference than strictly necessary.

To reiterate a point from my previous reviews, this is a first-order one dimensional modelling study that has very broad applicability to evolutionary biology and ecology, and that is a major advantage of this work. I think this broad relevance still gets lost in (quite a lot of) unnecessary introduction/discussion about the Neoproterozoic/Cryogenian and Palaeozoic/Cambrian.

A point that emphasises this is that my immediate thought on reaching line 94 (“Physiological stress is a potential driver of the emergence of evolutionary innovations...”) was “here is the start of the introduction”. The majority of the essential information for the introduction is in lines 94 to 151, with then one or possibly two paragraphs of Neoproterozoic-Phanerozoic transition context at line 140. Setting the work up in this way would help readers understand that this has broader applicability and that an interesting application of this work is to the Proterozoic-Phanerozoic transition. A similar point applies to the current framing of the discussion which could be reorganised to lead with and emphasise the general results of the study and then discuss what this would mean in the case of the Proterozoic-Phanerozoic transition.

Some specific points (using the line numbers in the merged pdf 508067_2_merged_1731277702.pdf):

Lines 6-7: "... the mechanisms linking these environmental conditions to physiology and then early organismal ecology are unclear." Unless making a very general point that organisms' (animals? see next comment) physiology provides the mechanistic link between environment and ecology, in the context of this study I understand this point to imply an strong environmental control on early animal diversification, which is an unproven assumption; if it's my mistaken assumption then apologies for that. I would suggest rephrasing this sentence to be more circumspect about whether there was an environmental control. This is something the authors do actually go on to show a mechanism for.

Lines 6-7: "... the mechanisms linking these environmental conditions to physiology and then early organismal ecology are unclear." Do the authors here mean "organismal" or "animal", in the context of their study?

Line 47: "temporal extent of potentially global glaciations of the Cryogenian"? The geographic extent of Cryogenian glacial conditions is not a settled topic.

Line 47: "(blue vertical fields)"? To distinguish from the blue colour in the temperature profile.

Line 51: "surface air temperature"?

Line 64: suggest deleting "alone"; its current inclusion suggest that there was a greater than four-fold increase in shallow settings globally, which may or may not be true but can't be inferred from one continent alone. It would be okay to keep if the authors referred to an area rather than a relative proportion.

Line 67: suggest also citing Peters & Gaines (currently ref 23) as they also drew the link with Sr isotopes, and consider also mentioning Nd isotopes which may tell a similar story.

Line 81-2: (a) if including oxygen isotope evidence, I would suggest including reference to Wotte et al. (2019) as well; (b) for modelling, I would suggest referencing Nardin et al. (2011) who modelled late Cambrian climate (reference on line 79 is to the whole Cambrian Period) and at least one of Scotese et al. (2021) or Valdes et al., (2020) who modelled Phanerozoic-scale temperatures including the Cambrian; (c) I would suggest also citing Wong Hearing et al. (2021) (already ref 37) here because reference is made to both climate modelling and climatically sensitive lithologies.

Line 113: please spell out "PAS" at first use.

Line 118: what different points are citations 43 and 46 referencing here? (i.e. why are they split around "(Fig. 1C)"?)

Fig 2: much better. Please make sure all abbreviations are spelled out in the caption.

Lines 150-1: stick with one or the other of "and"/"or" for "effective and poor"/("eOSM or pOSM").

Line 156: "interface"

Line 161: Following the points above about this being a general study, not need to reference Cambrian daylength in the caption.

Line 182: purely personal preference, but I'd suggest "day" and "night" rather than "daytime" and "nighttime" here and elsewhere.

Line 192-5: I appreciate the removal of "Cryogenian" and "Cambrian", but I would also avoid using "icehouse" and "greenhouse" here. I would suggest sticking to cold/cool/warm/hot as this could also reflect latitudinally different temperatures. The very generally applicable modelling work could be used to infer something about e.g. different locations in the modern ocean, like the difference between a polar setting with a steep bathymetric gradient and tropical setting with a wide shallow sea.

Line 199: consider a different descriptor (e.g. "hot" rather than "warm") for the 40 C runs. There is a lot of difference between a 25 C ocean and a 40 C ocean, and one might expect major mortality at 40 C.

Fig 3: I again ask the authors why they have not included the 40 C simulations as z-t-O₂ space plots?

Fig 3E: I make one last attempt to ask the authors to clarify Fig 3E. As I said before, the y-axis scales are very different, the dashed versus solid outlines are almost impossible to differentiate, and although it's slightly more intuitive with one colour abutting each axis, it is still not explicit which axis applies to which symbol set or colour.

Line 235-240: these sentences are purely discussion, not results, and are not needed here.

Line 273-4: do the authors mean "growth of an individual" or "increase in population size"? Not entirely clear at present.

Line 354-377: from "which could be, for example, ..." is pure discussion material. I would suggest moving all of it to the Discussion.

Line 419: I think "+/-" symbol in this context is "approximately" rather than "positive 5 or negative 5 degrees C"? Suggest replacing with "~" or "about".

Line 465: all organisms alive now will surely "use refined mechanisms"; surely here the meaning is that

Line 496-7: first clause needs referencing.

Line 511: I don't quite follow the grammar here – how does higher productivity offer large areas?

Line 526: grammar of the inserted text doesn't quite work.

Line 533: the climate of the Ediacaran Period is still rather uncertain. Depending on which lines of evidence you follow, there may have been one long (Wang et al., 2023a, 2023b) or multiple shorter (Linnemann et al., 2022; Niu et al., 2024) glaciations in the late Ediacaran. What the climate of the early Ediacaran was like is particularly uncertain.

Line 580-4: This is a great point and is worthy of a bit more here.

Fig 5: something odd with the colouring of the mid-Neoproterozoic panel – blue for sky and for water?

Fig. S1: this is still quite hard to follow, though removing the red has helped. Again, I would suggest simplifying this, or making it a multi-panel plot.

References cited:

Linnemann, U., Hofmann, M., Gärtner, A., Gärtner, J., Zieger, J., Krause, R., Haanel, R., Mende, K., Ovtcharova, M., Schaltegger, U., Vickers-Rich, P., 2022. An Upper Ediacaran Glacial Period in Cadomia: the Granville tillite (Armorican Massif) – sedimentology, geochronology and provenance. *Geol. Mag.* 159, 999–1013.
<https://doi.org/10.1017/S0016756821001011>

Nardin, E., Godd ris, Y., Donnadi u, Y., Hir, G.L., Blakey, R.C., Puc at, E., Aretz, M., 2011. Modeling the early Paleozoic long-term climatic trend. *Geol. Soc. Am. Bull.* 123, 1181–1192. <https://doi.org/10.1130/B30364.1>

Niu, Y., Shi, G.R., Zhang, Q., Jones, B.G., Wang, X., Zhao, G., 2024. Ediacaran Cordilleran-type mountain ice sheets and their erosion effects. *Earth-Sci. Rev.* 249, 104671. <https://doi.org/10.1016/j.earscirev.2023.104671>

Scotese, C.R., Song, H., Mills, B.J.W., van der Meer, D.G., 2021. Phanerozoic paleotemperatures: The earth's changing climate during the last 540 million years. *Earth-Sci. Rev.* 215, 103503. <https://doi.org/10.1016/j.earscirev.2021.103503>

Valdes, P.J., Scotese, C.R., Lunt, D.J., 2020. Deep Ocean Temperatures through Time. *Clim. Past Discuss.* 1–37. <https://doi.org/10.5194/cp-2020-83>

Wang, R., Shen, B., Lang, X., Wen, B., Mitchell, R.N., Ma, H., Yin, Z., Peng, Y., Liu, Y., Zhou, C., 2023a. A Great late Ediacaran ice age. *Natl. Sci. Rev.* nwad117. <https://doi.org/10.1093/nsr/nwad117>

Wang, R., Yin, Z., Shen, B., 2023b. A late Ediacaran ice age: The key node in the Earth system evolution. *Earth-Sci. Rev.* 247, 104610. <https://doi.org/10.1016/j.earscirev.2023.104610>

Wong Hearing, T.W., Pohl, A., Williams, M., Donnadieu, Y., Harvey, T.H.P., Scotese, C.R., Sepulchre, P., Franc, A., Vandenbroucke, T.R.A., 2021. Quantitative comparison of geological data and model simulations constrains early Cambrian geography and climate. *Nat. Commun.* 12, 3868. <https://doi.org/10.1038/s41467-021-24141-5>

Wotte, T., Skovsted, C.B., Whitehouse, M.J., Kouchinsky, A., 2019. Isotopic evidence for temperate oceans during the Cambrian Explosion. *Sci. Rep.* 9, 6330. <https://doi.org/10.1038/s41598-019-42719-4>

Point by Point Response to Reviewers' comments

Reviewer #3 (Remarks to the Author):

I am honestly a little disappointed to be reviewing this manuscript for the fourth time and finding the authors disregard my suggestions yet again. I really am just trying to help the authors anchor an interesting modelling approach in a representative and up-to-date reading of the Ediacaran-Cambrian literature. I have outlined many suggestions about how the Ediacaran biota relate to eumetazoan clades in previous reviews, so I won't repeat them here. I suggest a reframe of the discussion to avoid making poorly supported overstatements about a driver of the Cambrian explosion and instead describe an interesting mechanism that may have impacted animal diversification at some point between the Cryogenian and Cambrian.

We appreciate the very direct review that made us see the Reviewer's points in a new and clearer light. We also appreciate the reviewer comments that have been indeed helpful as we now have fully absorbed them. In response, we have now put more effort into mediating, merging, and balancing the work in the direction stressed by the reviewer, downscaled the (over)statements and updated the evolutionary framework and added new literature citations. Indeed, by absorbing more current literature from the Neoproterozoic community, it seems now clearer that several of our statements were too generalized, out of date, and without enough nuance. We hope the new text reflects a more educated and humble tone.

We have taken the changes as follows (plus smaller edits in additional instances):

- Changed in title, abstract (row 14, 28, 37, 45 etc) and throughout to capture a possible mechanism in a wider time interval, e.g., end-Neoproterozoic and early Paleozoic or Neoproterozoic-Paleozoic boundary.*
- Edited to 'Ediacaran and early Paleozoic' instead of Cambrian (row 500).*
- Added 'Ediacaran and Cambrian periods' (row 501).*
- We edited to clarify that cnidaria are represented in the Ediacaran (row 545).*
- Changed to "energy changed with the Neoproterozoic to early Palaeozoic flooding events" (row 570).*
- Removed that 'physical, functional, and ecological factors align' (row 573).*
- Changed to "Previous explorations into the animal diversification in the Neoproterozoic-early Palaeozoic" (row 601).*
- Changed to "Our results suggest that severe daily redox variations in an expanding benthic shallow shelf created niches in the end-Neoproterozoic and early Palaeozoic that could have promoted adaptive radiation (row 614).*

As per my previous reviews, I remain critical of the authors' overly confident descriptions of Cambrian vs Cryogenian climate and evolutionary trajectories. All that I think is really required here is a broader acknowledgment and appreciation that these mechanisms may have been important at some point between the Cryogenian and Cambrian but that we are not currently able to reconcile exactly when. I think that uncertainty is fine in the context of an interesting modelling study and don't really understand why the authors continue to try to overinterpret their results in an outdated

evolutionary and climatic framework here.

Yes, we agree (now). With a humbled perspective on the ‘long fuses’ to animal diversification from within the Neoproterozoic, we agree with the reviewer that a more modest claim that the proposed mechanisms may have been important and with impact, therein are both fair and more scientifically sustainable. We have fully re-drawn the phylogenetic tree (although simplified) and its divergence estimate in Figure 1 and specifically in panel 4F. The phylogeny of animal lineages is based on (Philippe, Brinkmann et al. 2011), with approximate divergence between lineages based on (Telford, Budd et al. 2015) and (Dunn, Liu et al. 2021). Although their estimates differ, we place the total group Metazoa at 574-600 Ma or mid-Ediacaran, while Cnidarian is placed at 574 Ma from (Dunn, Liu et al. 2021).

This comment indeed made us realize and now emphasize that components of the HIF system also evolved within animal groups, such as the Protostomia . The implications of fully absorbing this input are (as the reviewer likely saw long before us), that pOSM vs eOSMs could have been found both within and between animal groups. If OSM evolved within, e.g., Cnidarian, the range of adaptations would likely have differed within Ediacaran cnidarians as well as between, e.g., non-Ediacaran and Ediacaran cnidarians. Therefore, we now map and further discuss extant differences of HIF components between and within animal groups. We have made recovered the sequences used in Mills et al., 2018 and Graham and Presell 2017 to compare. Of course, the annotations have changed some so a few components – such as P564 in Nematostella and Trichoplax – are indeed less certain today (see differences in the new Table S4). However, an increase more complex combinations across the Metazoa appears with particular high diversity in the species tested from within Protostomia (see new phylogeny in Figure 4F). We think this comment significantly forwarded the discussion on how these many components to the OSM may have been adaptations to changes in the shelf environment. We have also changed the text, such as on row 354:

“This could separate Placozoa (pOSM) from Cnidaria (eOSM), or pOSM from eOSM species within cnidarians or bilaterians in both the Ediacaran (Dunn, Liu et al. 2021) or Cambrian (Fig. 4F).”

Humbled by the experience of not fully knowing the Neoproterozoic literature and phylogenetic studies of the Ediacaran biota, we decided not to venture out into comparisons of between e.g., Charnia or Nematostella or Dickinsonia, Kimberella, or Nematoda. However, based on how we know that the different components to the OSM of humans (e.g., CTAD, NTAD, FIH, P402, P564) lead to different sensing and responding to O₂ fluctuations, we can presume that animals with slightly different compositions (whether Charnia or Nematostella) had slightly different OSMs. Or, similarly, that bilaterians like Dickinsonia, Kimberella, or Nematoda may have had as different OSM as we see between extant protostomes. But again, in the current manuscript, we will avoid opening another discussion on the details of Ediacaran organisms other than paying them respect as Metazoa.

I agree with the similar comments of reviewer 1 and really struggle to understand why we have gone through all of these rounds of review when the authors simply could have

taken our suggestions on board. I can't help but feel that it is a waste of everyone's time to be debating this in rebuttal letters when the authors could have incorporated our referenced suggestions with ease. New analyses may be reasonably out of scope, but a discussion reframe would have been straightforward for the authors to incorporate at any juncture.

Again, we acknowledge this point and believe that we have finally taken these very valuable suggestions onboard in this current version, include those described above and in the thorough rewrite of the manus. We apologies for rounds of review and appreciate the reviewer's patience.

In its current form, I cannot recommend this paper for publication in Nature Communications. As emphasized repeatedly in my multiple reviews, fixing these fundamental issues really wouldn't be that hard but (unfortunately) the authors don't seem willing to do so.

Yes, we are willing to and have now had the bandwidth to improve the manuscript. Thanks for the frank wake-up comment.

Line 37 – I don't see the relevance of the reference to cancer here and strongly suggest removing.

Yes, we can see this is distracting. Removed.

Line 61 – interpretation of Sr isotope records is contentious. They certainly don't "demonstrate" this. I suggest you qualify the statement or remove.

Thanks, and we agree. We have edited to let the estimate acquired via Macrostrat 'demonstrate' the increase, as? this would only be consistent with the Sr record.

Line 86 – "coupled to" – correlated with? A potential driver of?

Thanks yes. Now changed to 'a potential driver of'.

Line 130 – specifically in the Cambrian? Why not anchor your hypotheses on this broad distinction between the Cryogenian and early Paleozoic. The more specific your time references here, the less compelling your narrative, in my opinion.

Yes, finally we see this point and are able to 'kill' the darling or narrow focus on the Cambrian. Changed to 'from the late Neoproterozoic to the early Paleozoic' throughout.

Line 326 – "we conjecture that oxygen sensing mechanisms evolved within early animals at the Cambrian flooding event" – I don't understand how you can invoke oxygen sensing evolution in early animals at ~520Ma (critical objection being the use of 'early'). Yet again, I feel like the discussion here is completely missing an informed perspective on the divergence times of early animals. Surely this is way too late for these higher level divergences to be associated with innovation in OSMs? If you disagree then please justify.

Yes, the reviewer is entirely correct that this is too late to fully innovate OSM. What we mean to say is that adaptations within the OSM may have been selected for. As mentioned above, we have redrawn the representations of animal phylogeny and divergence times (Figure 1 and 4F), which have forced us to re-evaluate a previously naïve view on the changes of OSM. To the extent that components of OSMs are studied

outside of humans, it is clear that the function and differences are much more complicated than just having HIF-1a or not. It is also more complicated that some animal groups have specific OSM components within their group. The fine detail to how OSM components differ within animals groups, but also increases its complexity from sponges to vertebrates is now visible in Figure 4F. With this in mind, it is also clear that the many components to OSMs (e.g., CTAD, P564, NTAD, FIH) in extant animals each fill fundamental roles in the sensitivity and response to chemical variations (in internal and external environments). We have therefore rewritten the entire paragraph from row 353 (and at instances in the text above):

“The population and phenotype model is agnostic to what organisms would represent p The population and phenotype model is agnostic to which organisms would represent pOSM or eOSM species, which could be, for example, Porifera vs Cnidaria, Cnidaria vs Bilateria, or invertebrates vs vertebrates within Bilateria. Here, we suggest that adaptations of oxygen sensing mechanisms can have been selected for within animals at events that led to specifically periodically fluctuating redox conditions (Fig. 4D-E). While HIFs that today compose the OSM of modern animals contain components (e.g., genes, domains, or proteins) that can be traced back to the origins of yeast and even to prokaryotes (Hammarlund, Flashman et al. 2020 and references therein), its building blocks expand within and between early branching animals, including poriferans, cnidarian, or molluscs. Oxygen sensing in animals with e.g., only P564 would be simpler than in animals e.g., with P564 and P402 or with P564 and CTAD. This could separate Placozoa (pOSM) from Cnidaria (eOSM), or species with pOSM from eOSM within cnidarians or bilaterians; whether living in the Ediacaran (Dunn, Liu et al. 2021) or Cambrian (Fig. 4F). Components to the OSM like the P402, CTAD, NTAD and Factor Inhibiting HIF (FIH) regulate a wide scope of target genes at a wide range of conditions, some of them with clear importance for development (Hammarlund, Flashman et al. 2020). [...]”

Figure 4 – as I have said in every review of this paper, I firmly feel that the evolutionary timelines invoked for different metazoan lineages here is very out of date. For example, when we look at the Bilateria illustration in Fig 4F, what about the Ediacaran bilaterians? Dickinsonia, Spriggina and bilaterian trace makers are all fully disregarded here so far as I can tell. Similar stories can be applied to your Cnidaria and Porifera extents.

Yes. We now agree with this point and see how others have painstakingly mapped animal evolution in the deeper Neoproterozoic. Indeed, the acknowledgement of work that places total-group Eumetazoa into the mid-Cryogenian (Philippe, Derelle et al. 2009, Telford, Budd et al. 2015, Dunn, Liu et al. 2021) forces us to view the differences in oxygen sensing mechanisms, not only between Cnidaria and Bilateria, but also between Protostomia and Deuterostomia. A more careful mapping of the diversity of OSM also within Bilateria (using e.g., Mills and Canfield 2014, Graham and Presnell 2017, Song, Modjewski et al. 2022) not only makes us realize that the previous representation in panels E-F were too general and simplistic. Instead, the comment and reading certainly makes it so clear that the argument and mechanism we wish to put forward are those of

selection, adaptation and diversification. See also Table S4 for comparison of what these analyses gave in 2017, 2018, and today.

In response:

- We have re-made panel E and F. In E, we've only depicted the conditions that we argue contribute to the mechanism at the PC/C boundary (dashed). We aim for this dashed box to fill emphasize that we by no means make claims to accurately time the Neoproterozoic roots of animal evolution.
- We depict the simplified phylogeny of animal evolution in Panel F. This phylogeny stretches into the Neoproterozoic (of panel D and in Figure 1) but without an adjacent exact time line. The nodes are built on Telford et al., 2015 and with nodes placed in general terms from the Dunn et al., 2021 paper. We feel the updated Panel F serves two purposes: to make a more current representation of the deep roots of metazoan lineages plus that the diversity of OSMs are much more than just having HIF-a or not.

Lines 482 onwards – refer to above and previous comments on how out of date I feel this discussion of different lineages is. I strongly recommend that this would have to be updated to represent current understanding of the history of these lineages in order for this manuscript to be published in this form. I have provided detailed arguments on this in previous reviews and really don't understand why the authors are so resistant to doing this. If they fundamentally disagree with the literature that I cite then it would be useful to justify this in the text and put forward an argument that is neutral to these reconstructions. Currently this out of date discussion just weakens a perfectly good modelling study in my opinion, and will only frustrate readers in the Neoproterozoic community. Having spent the time to outline these arguments to the authors multiple times between my four reviews of this manuscript, I honestly am quite frustrated that they have not taken the small amount of time it would take to incorporate this more up to date discussion into their narrative.

Again, we appreciate the patience of the reviewer to prod us to pay attention to the advancements in the Neoproterozoic literature. After reading this literature better, we fully agree and have gone over the manuscript with a new lens at hand. Apart from changes to "late Neoproterozoic-early Paleozoic" throughout (see above), we also add notes to the redrawn phylogeny and expansion of pOSM and eOSM species in both the Ediacaran and Paleozoic:

- Oxygen sensing in animals with e.g., only P564 would be simpler than in animals e.g., with P564 and P402 or with P564 and CTAD. This could separate Placozoa (pOSM) from Cnidaria (eOSM), or species with pOSM from eOSM within cnidarians or bilaterians; whether living in the Ediacaran (Dunn, Liu et al. 2021) or Cambrian (**Fig. 4F**).
- The divergence of animal clades (one of several possible time-calibrated trees, based on (Philippe, Brinkmann et al. 2011), with approximate divergence between lineages based on (Telford, Budd et al. 2015) and (Dunn, Liu et al. 2021)). Boxes represent early branching clades and Bilateria that today perform cellular oxygen sensing mechanisms with the Hypoxia-Inducible Factor (HIF) system (filled orange) or not (grey), based on (Graham and Presnell 2017, Mills, Francis et al. 2018). The positions of Ctenophora and Placozoa are based on (Philippe, Brinkmann et al. 2011).

Line 511 – “time, a cascade of physical, functional, and ecological factors aligns directly with the Cambrian Period.” I would argue this is patently untrue. Why overreach like this? There is a cool underexplored mechanism here that you have the opportunity to tell the reader about. Why frustrate the reader by making claims like this that are clearly dismissed by a quite superficial understanding of the literature?

Thanks yes, we see this now. This is now changed to “The underexplored mechanism for the selection of a cellular machinery to handle redox fluctuations can be inferred within the framework of physical, functional, and ecological changes in the end-Neoproterozoic and early Palaeozoic.”

Fig 5 caption – see notes above.

Also here, the text and labels in the figure are updated. Thanks for these comments.

Reviewer #4 (Remarks to the Author):

The authors have done a reasonable job of addressing the substantive points from my previous reviews. I would be happy to see this manuscript published following a few minor revisions which I still think are necessary and which I hope the authors find helpful.

We are grateful to the reviewer’s comments here and previously. We have addressed these comments in full, as explained below.

There are some line-by-line comments below, but first I would strongly encourage the authors to fully revise the text to separate their general, first order, modelling results from their interpretation of how this would play out in the Neoproterozoic-Phanerozoic transition. In particular, this is very generally applicable, first order modelling. At various places, the text could be clarified to show that:

a. The first order modelling of oxygen concentration cycles was done for a range of T, TOC, and [O₂] conditions which could be interpreted broadly in a geographical and temporal way (e.g. for various places on present-day Earth or times in the geological past). Keeping this distinction clear throughout would help emphasize the broader applicability of the concepts modelled here and would widen the appeal of the study.

We agree and have made an effort to separate results from interpretations as well as making them time-agnostic. For example, in the section on the biogeochemical model and results, we made the following changes:

- *Row 166: Removed ‘Cambrian’ for a later reference to ‘(early Paleozoic)’.*
- *All references to Cryogenian/Cambrian climate are replaced with cold or warm.*
- *Rephrased the text to better describe the basic functionality of the biogeochemical model used to quantify sediment O₂ dynamics (l. 165-186).*
- *Rephrased the text to make the immediate modelling results more clear in a non-interpreted way based on the basic biophysical responses (l. 188-238), e.g. (l. 188) “These first-order experiments, we found that in an icehouse scenario, daily oxygen fluctuations and diel amplitude would have been modest with weak oxic conditions during the day and with hypoxic conditions during night lasting for ~11*

h (Fig. 3A, 3C). In the historically later greenhouse scenario, daily fluctuations were profoundly amplified and would have demonstrated abrupt changes from fully oxygenated conditions during daytime to true anoxia during nighttime in less than 0.3 h (Fig. 3B, 3D)."

- *Edited to row 412: "By applying two lines of first-order reconstructions"*.

b. The first order modelling of pOSM and eOSM species could also be shown as of much broader relevance than it is currently presented as. In particular, the authors acknowledge that there are different differences in OSM efficiency between different animal clades (bilaterians, cnidarians, placozoans, poriferans, ctenophores). The focus in the results section on presenting this work as predominantly about the differences between bilaterians and other animals hides the wider implications which could be seen more broadly in modern and other paleo-ecological settings. To keep to the early animal example, such an approach could be applied to the redox oscillations (on a different time scale, but same principle) of the later Ediacaran world, rather than just the Cambrian.

We fully agree and the comments from both the reviewers have encouraged us to revise that previously simplistic view. Indeed, the figure panel 4F is re-drawn to emphasize how much the components of OSM differ between, but also within, animal groups. The text is also changed to capture this. For example, at row 343, we summarize the results with the following changes to the text:

"The population and phenotype model is agnostic to which organisms would represent pOSM or eOSM species, which could be, for example, Porifera vs Cnidaria, Cnidaria vs Bilateria, or invertebrates vs vertebrates within Bilateria. Here, we suggest that adaptations of oxygen sensing mechanisms can have been selected for within animals at events that led to specifically periodically fluctuating redox conditions (Fig. 4D-E). While HIFs that today compose the OSM of modern animals contain components (e.g., genes, domains, or proteins) that can be traced back to the origins of yeast and even to prokaryotes⁴¹ and references therein, its building blocks expand within and between early branching animals, including poriferans, cnidarian, or molluscs. Oxygen sensing in animals with e.g., only P564 would be simpler than in animals e.g., with P564 and P402 or with P564 and CTAD. This could separate Placozoa (pOSM) from Cnidaria (eOSM), or species with pOSM from eOSM within cnidarians or bilaterians; whether living in the Ediacaran¹⁷ or Cambrian (Fig. 4F). Components to the OSM like the P402, CTAD, NTAD and Factor Inhibiting HIF (FIH) regulate a wide scope of target genes at a wide range of conditions, some of them with clear importance for development⁴¹."

Finally, I do hope to see this published soon and I think the authors are very nearly there with it.

Much appreciated, thanks so much for all the input.

Line-by-line comments:

Line 9: Proterozoic-Phanerozoic, or Neoproterozoic-Paleozoic, would be preferable to Precambrian-Cambrian.

We agree and have changed to Neoproterozoic-Paleozoic here, plus edited 'PC/C' also onwards in the manuscript.

Lines 27 to 28: ditto Line 9.

Thanks, yes. Edited here and throughout the manuscript.

Fig. 1: Please define the dashed box label "ROI".

Defined.

Line 51 (Fig. 1 caption): rephrase to make one sentence about the blue-red transition; it doesn't currently flow. E.g. "+25 °C marking a transition from icehouse to greenhouse climate states". Otherwise, the caption is much improved, thanks.

Edited.

Lines 72 to 74: climatically sensitive lithologies have also contributed substantially to this picture (e.g. Boucot, A.J., Xu, C., Scotese, C.R. and Morley, R.J. 2013. Phanerozoic Paleoclimate: An Atlas of Lithologic Indicators of Climate, 1st ed. SEPM Concepts in Sedimentology and Paleontology 11).

Thanks, yes. Now added.

Lines 74 to 76: something has gone wrong here, I think. Please check for grammar and punctuation.

Indeed, yes. The sentence is now changed.

Line 109: "ctenophores" or "Ctenophora".

Edited.

Line 117: this paragraph is good but, as someone who is not a biologist or biochemist, I only follow the implications of it through having already reviewed the paper. At Line 117, you could be a bit more explicit that each of the additions to the HIFa pathway increased its efficacy and so represent step-wise acquisitions of O₂ sensitivity across the animal tree.

Thanks, we can see this now. We have edited the last part of the paragraph as follows:

"These animals share a joint core to this sensing by an oxygen dependent degradation domain (ODDD) of the bHLH-PAS protein where a proline can be targeted by a PHD protein (if O₂ is present) and hinder gene transcription (see also Supplementary Information). However, the components of the pathway vary in their presence within animal groups and in terms of how they sense and respond to oxygen fluctuations. For example, an additional C-terminal transactivation domain (CTAD) can enable the protein to bind with transcriptional coactivators of and a factor inhibiting HIF (FIH) can limit the regulation of genes.²¹ Differences in oxygen sensitivity and regulatory roles are also described for the two different prolines (P564 and P402) targeted by PHD, and an N-terminal transactivation domain (NTAD).²¹"

Fig. 2: I reiterate that the maps in A and B should have the same colour scales – they are representing different temperatures of water mass, but still water mass. At present, this

looks more like topography than flooded continental area and mis-represents that the continents were not completely flooded in the Cambrian. I still think the maps add confusion as this is not a latitude, or even geography, based study, but more about flooded continental shelf. The onshore-offshore profiles show this better. This is particularly pertinent as the next section talks immediately about a 1D model, no horizontal (map) resolution needed.

We concur and have removed the upper visualization of the flooded continents.

Line 168: “or anoxia” is superfluous. Just need “O₂ concentration.” As levels oxygenation are defined below (lines 172 to 174).

Thanks, yes. This is edited now.

Lines 168 to 169: I don't follow the grammar of “Since quick O₂ changes require benthic animals to conform their ATP-turnover”. Should this be “conform to their ATP-turnover rate”? or something else?

Thanks, and we see the confusion. Indeed, we meant to say (and have now changed the sentence to): “Since quick O₂ changes can be physiologically stressful to benthic animals where adjustments to their ATP-turnover are less quick, we also measure the transition time between day- and nighttime conditions.”

Lines 176 to 178: I thought from their last response that the authors were refraining from calling the scenarios “Cryogenian” and “Cambrian” and instead sticking to “cold/warm/hot” or “icehouse/greenhouse” and then interpreting these later as reflecting Cryogenian and Cambrian. Presenting it in this way would emphasize that this is a first-order simulation experiment, not an actualistic one, particularly as some Cryogenian intervals were quite possibly greenhouse or super-greenhouse climate states. This would also emphasise the broader applicability of the work: this is just looking at pO₂ variation under different T, TOC, [O₂] conditions – these apply in the modern world (e.g. by ocean basin, by latitude) and in numerous other deep-time contexts. The generality of this is a strength that could be much better shown.

Yes, we completely agree (now) and have edited throughout the manuscript to avoid these premature associations. Here in this part, all Cryogenian or Cambrian are stricken.

Fig. 3: I again urge the authors to make this figure easier to follow and assess. It should be split up or made bigger, or something. I also suggest again plotting the blue and yellow bands in Fig 3E either on the same scale or on separate plots: their numeric range is two orders of magnitude different and it is still not obvious without the caption which coloured envelope relates to which y-axis scale, particularly as the same blue and yellow are used for different oxygen concentrations in figs 3A & 3B.

We see the reviewer's point and we have therefore:

- *Changed the color coding to be the same as in the A-B modeling results.*
- *Moved each field closer to its y-axis to clarify.*
- *Emphasized in the caption that the data is primarily represented by the symbols (not the fields).*
- *We have increased font size of labels to 16 pts.*

Line 319: “ $d \geq 0.01$ ” includes “ $d \geq 0.09$ ”. I think a “, respectively” is needed to clarify that these values relate to stochastic and periodic fluctuations, respectively?

Thanks for the sharp eye that in fact made us see a mistake in the interpretation of the competition experiment results (previously also noted by the reviewer as confusing). Therefore, we have now revisited the results and both changed and expanded on this discussion. As seen in Fig. S8, it is clear that eOSM can hold the cost of a more efficient oxygen sensing mechanism in specifically environments with periodic and stochastic fluctuations. In environments with only period fluctuations, eOSM can carry a fairly high cost (until 0.08). In contrast, in environments with only stochastic fluctuations, the cost of an efficient oxygen sensing mechanism is soon too high and would favor pOSM. We are very grateful for the poke to revisit this experiment, since it emphasizes the potential role for period fluctuations to have another evolutionary force than the stochastic fluctuations that we have discussed more in the field (e.g., effects of upwelling etc). Changes in short:

- *Row 329-335 is rewritten.*
- *In the caption of the modeling results, we emphasize that a total of 33 competition simulations were run, across three environmental conditions and eleven different costs of the oxygen sensing mechanism (d) and that – for each of these simulations – the competition experiment is run until one of the species goes extinct while the other remains extant.*
- *We have updated the Figure S6, so that periodic fluctuations is denoted with an X instead, and is therefore visible also when below the circle.*
- *We have also added the code for these competition experiments in SI.*

Line 332: “cnidarians” (lowercase “c”) or “Cnidaria”.

Thanks, this is now edited together with the sentence.

Lines 326 to 342: here is an opportunity to illustrate the impact of stepwise acquisition of increasingly ‘e’OSMs, with each clade being slightly more ‘e’ than those earlier branching clades. I think I made a similar point before, but here the key is not the ‘bilaterians are best’ but that each clade made some slight improvements in their OSM. The text of this paragraph could again be strengthened by saving until the end of the paragraph the specific case of early animal evolution through the Neoproterozoic-Phanerozoic transition and keeping the initial interpretation as hypothetical discussion of organisms with less (‘p’) or more (‘e’) efficient oxygen sensing mechanisms.

We are grateful for this view and comment. We have now added to the paragraph after modeling results:

“The population and phenotype model is agnostic to what organisms would represent pOSM or eOSM species. The pOSM vs eOSM could be, for example, Porifera vs Cnidaria, Cnidaria vs Bilateria, invertebrate bilaterian vs vertebrates, or any combination within animal groups.”

We also edited that “adaptations of rudimentary oxygen-sensing capacities gave certain animals a competitive advantage”. We have also updated Figure 4E with a more detailed schematic of the differences within OSM of animals.

Fig. 4: This is a lot better, and I really like the addition of panel F. I do again ask about the colour variation in the diel bar – are we to read anything into the yellow-grey-green

transitions and their placement?

We are grateful to the reviewer pointing this out again; a point that we know has remained important, but hard to visualize. The reviewer's point made us rotate the direction of the color variation 90 degrees. The distinct color, aimed to visualize that daily benthic O₂ fluctuations were severe, is now stretching over the entire time frame that we define as our region of interest (ROI). A contrasting color bar for modest daily benthic oxygen fluctuations (imagine less distinct color differences) would likely bring this point home more clearly, especially if placed in at the end of the Cryogenian. However, we hesitate to expand indications for both temperatures or metabolic energy outside of our ROI since it adds uncertainty. We think the least uncertain assumptions, that global temperatures were lower at the end of the Cryogenian, is clear enough from Figure 1. We hope that you agree.

Fig. S1: this is really hard to interpret, particularly for anyone who has difficulty with their colour vision. I would urge the authors again to simplify this and use colour palettes that are robust to common colour vision deficiencies.

Yes. We have now changed the color of the Mills et al., record to just dark gray (removed our coloring). Also, we added direct links to both manuscripts (Mills et al., and also Bergmann et al.,) so that comparison is easier for the reader.

References

Dunn, F. S., et al. (2021). "The developmental biology of *Charnia* and the eumetazoan affinity of the Ediacaran rangeomorphs." *Science Advances* **7**(30): eabe0291.

Rangeomorphs represent a long-extinct group of eumetazoans with a bodyplan unlike anything alive today. Molecular timescales estimate that early animal lineages diverged tens of millions of years before their earliest unequivocal fossil evidence. The Ediacaran macrobiota (~574 to 538 million years ago) are largely eschewed from this debate, primarily due to their extreme phylogenetic uncertainty, but remain germane. We characterize the development of *Charnia masoni* and establish the affinity of rangeomorphs, among the oldest and most enigmatic components of the Ediacaran macrobiota. We provide the first direct evidence for the internal interconnected nature of rangeomorphs and show that *Charnia* was constructed of repeated branches that derived successively from pre-existing branches. We find homology and rationalize morphogenesis between disparate rangeomorph taxa, before producing a phylogenetic analysis, resolving *Charnia* as a stem-eumetazoan and expanding the anatomical disparity of that group to include a long-extinct bodyplan. These data bring competing records of early animal evolution into closer agreement, reformulating our understanding of the evolutionary emergence of animal bodyplans.

Graham, A. M. and J. S. Presnell (2017). "Hypoxia Inducible Factor (HIF) transcription factor family expansion, diversification, divergence and selection in eukaryotes." *PLoS one* **12**(6): e0179545.

Hammarlund, E. U., et al. (2020). "Oxygen-sensing mechanisms across eukaryotic kingdoms and their roles in complex multicellularity." *Science* **370**(6515).

Oxygen-sensing mechanisms of eukaryotic multicellular organisms coordinate hypoxic cellular responses in a spatiotemporal manner. Although this capacity partly allows animals and plants to acutely adapt to oxygen deprivation, its functional and historical roots in hypoxia emphasize a broader evolutionary role. For multicellular life-forms that persist in settings with variable oxygen concentrations, the capacity to perceive and modulate responses in and between cells is pivotal. Animals and higher plants represent the most complex life-forms that ever diversified on Earth, and their oxygen-sensing mechanisms demonstrate convergent evolution from a functional perspective. Exploring oxygen-sensing mechanisms across eukaryotic kingdoms can inform us on biological innovations to harness ever-changing oxygen availability at the dawn of complex life and its utilization for their organismal development.

Mills, D. B. and D. E. Canfield (2014). "Oxygen and animal evolution: Did a rise of atmospheric oxygen "trigger" the origin of animals?" *Bioessays* **36**(12): 1145-1155.

Mills, D. B., et al. (2018). "The last common ancestor of animals lacked the HIF pathway and respired in low-oxygen environments." *eLife* **7**: e31176.

Animals have a carefully orchestrated relationship with oxygen. When exposed to low environmental oxygen concentrations, and during periods of increased energy expenditure, animals maintain cellular oxygen homeostasis by enhancing internal

oxygen delivery, and by enabling the anaerobic production of ATP. These low-oxygen responses are thought to be controlled universally across animals by the hypoxia-inducible factor (HIF). We find, however, that sponge and ctenophore genomes lack key components of the HIF pathway. Since sponges and ctenophores are likely sister to all remaining animal phyla, the last common ancestor of extant animals likely lacked the HIF pathway as well. Laboratory experiments show that the marine sponge *Tethya wilhelma* maintains normal transcription under oxygen levels down to 0.25% of modern atmospheric saturation, the lowest levels we investigated, consistent with the predicted absence of HIF or any other HIF-like pathway. Thus, the last common ancestor of all living animals could have metabolized aerobically under very low environmental oxygen concentrations.

Philippe, H., et al. (2011). "Resolving Difficult Phylogenetic Questions: Why More Sequences Are Not Enough." *PLoS Biology* **9**(3): e1000602.

Philippe, H., et al. (2009). "Phylogenomics Revives Traditional Views on Deep Animal Relationships." *Current Biology* **19**(8): 706-712.

Summary The origin of many of the defining features of animal body plans, such as symmetry, nervous system, and the mesoderm, remains shrouded in mystery because of major uncertainty regarding the emergence order of the early branching taxa: the sponge groups, ctenophores, placozoans, cnidarians, and bilaterians. The "phylogenomic" approach [1] has recently provided a robust picture for intrabilaterian relationships 2, 3 but not yet for more early branching metazoan clades. We have assembled a comprehensive 128 gene data set including newly generated sequence data from ctenophores, cnidarians, and all four main sponge groups. The resulting phylogeny yields two significant conclusions reviving old views that have been challenged in the molecular era: (1) that the sponges (Porifera) are monophyletic and not paraphyletic as repeatedly proposed 4, 5, 6, 7, 8, 9, thus undermining the idea that ancestral metazoans had a sponge-like body plan; (2) that the most likely position for the ctenophores is together with the cnidarians in a "coelenterate" clade. The Porifera and the Placozoa branch basally with respect to a moderately supported "eumetazoan" clade containing the three taxa with nervous system and muscle cells (Cnidaria, Ctenophora, and Bilateria). This new phylogeny provides a stimulating framework for exploring the important changes that shaped the body plans of the early diverging phyla.

Song, B., et al. (2022). "The origin and distribution of the main oxygen sensing mechanism across metazoans." *Frontiers in Physiology* **13**.

Oxygen sensing mechanisms are essential for metazoans, their origin and evolution in the context of oxygen in Earth history are of interest. To trace the evolution of a main oxygen sensing mechanism among metazoans, the hypoxia induced factor, HIF, we investigated the phylogenetic distribution and phylogeny of 11 of its components across 566 eukaryote genomes. The HIF based oxygen sensing machinery in eukaryotes can be traced as far back as 800 million years (Ma) ago, likely to the last metazoan common ancestor (LMCA), and arose at a time when the atmospheric oxygen content corresponded roughly to the Pasteur point, or roughly 1% of present atmospheric level (PAL). By the time of the Cambrian explosion (541–

485 Ma) as oxygen levels started to approach those of the modern atmosphere, the HIF system with its key components HIF1 α , HIF1 β , PHD1, PHD4, FIH and VHL was well established across metazoan lineages. HIF1 α is more widely distributed and therefore may have evolved earlier than HIF2 α and HIF3 α , and HIF1 β is more widely distributed than HIF2 β in invertebrates. PHD1, PHD4, FIH, and VHL appear in all 13 metazoan phyla. The O₂ consuming enzymes of the pathway, PHDs and FIH, have a lower substrate affinity, K_m, for O₂ than terminal oxidases in the mitochondrial respiratory chain, in line with their function as an environmental signal to switch to anaerobic energy metabolic pathways. The ancient HIF system has been conserved and widespread during the period when metazoans evolved and diversified together with O₂ during Earth history.

Telford, Maximilian J., et al. (2015). "Phylogenomic Insights into Animal Evolution." Current Biology **25**(19): R876-R887.

Animals make up only a small fraction of the eukaryotic tree of life, yet, from our vantage point as members of the animal kingdom, the evolution of the bewildering diversity of animal forms is endlessly fascinating. In the century following the publication of Darwin's *Origin of Species*, hypotheses regarding the evolution of the major branches of the animal kingdom — their relationships to each other and the evolution of their body plans — was based on a consideration of the morphological and developmental characteristics of the different animal groups. This morphology-based approach had many successes but important aspects of the evolutionary tree remained disputed. In the past three decades, molecular data, most obviously primary sequences of DNA and proteins, have provided an estimate of animal phylogeny largely independent of the morphological evolution we would ultimately like to understand. The molecular tree that has evolved over the past three decades has drastically altered our view of animal phylogeny and many aspects of the tree are no longer contentious. The focus of molecular studies on relationships between animal groups means, however, that the discipline has become somewhat divorced from the underlying biology and from the morphological characteristics whose evolution we aim to understand. Here, we consider what we currently know of animal phylogeny; what aspects we are still uncertain about and what our improved understanding of animal phylogeny can tell us about the evolution of the great diversity of animal life.

Point by Point Response to Reviewers' comments

Reviewer #3 (Remarks to the Author):

Firstly, apologies to the authors for the delay in submitting my review at the end of a busy year. I am genuinely really pleased to see the changes that the authors have made in response to my comments and am glad that some directness seems to have paid off in this case. I think the increased generality and nuance here will (despite making broader claims) lead to a much more well received manuscript when the community have the opportunity to read it. Now that the authors have addressed so many of my key (hopefully constructive!) criticisms, I am happy to recommend this manuscript for publication in Nature Communications without any further rounds of review.

We are very grateful to the reviewers very constructive comments and continuous work to help us improve the manuscript. We are also happy that the reviewer considers the changes as sufficient and the current version suitable for publication. We agree it may be well received by the community.

Reviewer #4 (Remarks to the Author):

Firstly, the text is much clearer this time round, and I was pleased to read an improved version. Secondly, I don't have a problem with the fundamental science of this work (though I remain perplexed about why the results of the 40 C simulations are not really included), and I think it does have the broad relevance suitable for Nature Communications. Thirdly, however, I do still feel that there are improvements needed regarding how the work is contextualised and presented, but I think most of these are more personal preference than strictly necessary. To reiterate a point from my previous reviews, this is a first-order one dimensional modelling study that has very broad applicability to evolutionary biology and ecology, and that is a major advantage of this work. I think this broad relevance still gets lost in (quite a lot of) unnecessary introduction/discussion about the Neoproterozoic/Cryogenian and Palaeozoic/Cambrian. A point that emphasises this is that my immediate thought on reaching line 94 ("Physiological stress is a potential driver of the emergence of evolutionary innovations...") was "here is the start of the introduction". The majority of the essential information for the introduction is in lines 94 to 151, with then one or possibly two paragraphs of Neoproterozoic-Phanerozoic transition context at line 140. Setting the work up in this way would help readers understand that this has broader applicability and that an interesting application of this work is to the Proterozoic-Phanerozoic transition. A similar point applies to the current framing of the discussion which could be reorganised to lead with and emphasise the general results of the study and then discuss what this would mean in the case of the Proterozoic-Phanerozoic transition.

We are grateful for the effort and very constructive comments that the reviewer provides. Regarding the +40C scenario, we reply in full below. In essence, based on the step between +5C and the +25C step also represents the presumed increase in TOC

loading, we evaluate that a second step to +40C does not add much information or power to visualize the suggested change in temperature-driven redox fluctuations, see figure for comparison below. (The probability of all factors aligning to the suggested change in temperature-driven redox fluctuations is also evaluated in the Monte-Carlo simulation). In the now revised manuscript, we agree that the reviewers suggestions to e.g., remove and restructure text improved the structure. We describe these changes in detail below.

Some specific points (using the line numbers in the merged pdf 508067_2_merged_1731277702.pdf):

Lines 6-7: "... the mechanisms linking these environmental conditions to physiology and then early organismal ecology are unclear." Unless making a very general point that organisms' (animals? see next comment) physiology provides the mechanistic link between environment and ecology, in the context of this study I understand this point to imply an strong environmental control on early animal diversification, which is an unproven assumption; if it's my mistaken assumption then apologies for that. I would suggest rephrasing this sentence to be more circumspect about whether there was an environmental control. This is something the authors do actually go on to show a mechanism for.

Thanks yes, we can see that straw man now. We have rephrased this sentence to '*Animal diversification mirrors an expansion in marine shelf area under a greenhouse climate, though the extent to which these environmental conditions directly influenced physiology and early organismal ecology remains unclear.*'.

Lines 6-7: "... the mechanisms linking these environmental conditions to physiology and then early organismal ecology are unclear." Do the authors here mean "organismal" or "animal", in the context of their study?

Good point. With 'organismal', we mean to also include how also e.g., benthic microalgae were influenced by the expansion and, therefore, interacted with the ecology of animals and other organisms. Although the double meaning with organismal may not be entirely clear here in the abstract, we leave the sentence as is to possibly include also the other organisms.

Line 47: "temporal extent of potentially global glaciations of the Cryogenian"? The geographic extent of Cryogenian glacial conditions is not a settled topic.

Yes, changed.

Line 47: "(blue vertical fields)"? To distinguish from the blue colour in the temperature profile.

Yes, changed.

Line 51: “surface air temperature”?

Yes, changed.

Line 64: suggest deleting “alone”; its current inclusion suggest that there was a greater than four-fold increase in shallow settings globally, which may or may not be true but can’t be inferred from one continent alone. It would be okay to keep if the authors referred to an area rather than a relative proportion.

Agreed, changed.

Line 67: suggest also citing Peters & Gaines (currently ref 23) as they also drew the link with Sr isotopes, and consider also mentioning Nd isotopes which may tell a similar story.

Yes and thanks for the mention on Nd isotopes. Both Peters & Gaines 2012 and Wei et al., 2019 are added.

Line 81-2:

(a) if including oxygen isotope evidence, I would suggest including reference to Wotte et al. (2019) as well;

Agreed and added.

(b) for modelling, I would suggest referencing Nardin et al. (2011) who modelled late Cambrian climate (reference on line 79 is to the whole Cambrian Period) and at least one of Scotese et al. (2021) or Valdes et al., (2020) who modelled Phanerozoic-scale temperatures including the Cambrian;

Thanks for all three of these, where the Valdes et al was new to us and much appreciated. Nardin and Valdes are added.

(c) I would suggest also citing Wong Hearing et al. (2021) (already ref 37) here because reference is made to both climate modelling and climatically sensitive lithologies.

Agreed and added.

Line 113: please spell out “PAS” at first use.

Changed.

Line 118: what different points are citations 43 and 46 referencing here? (i.e. why are they split around “(Fig. 1C)”?)

Good point. The Semenza reference (44) refers to a first description of the role of HIF-1a for responding to variations in oxygen concentrations (although fairly unspecific) and the Loenarz reference (47) to a first mapping of HIF-1a across the animal clade. We have now moved them together. Changed.

Fig 2: much better. Please make sure all abbreviations are spelled out in the caption.

Yes, we have added explanations to the model parameters: Model parameters are atmospheric oxygen (ppO₂), temperature (T), salinity (S), production Q10 (Q10p), respiration Q10 (Q10r), and TOC adjustment factor (c_{oc}).

Lines 150-1: stick with one or the other of “and”/”or” for “effective and poor”/ (“eOSM or pOSM”).

Corrected now, to ‘or’.

Line 156: “interface”
Edited.

Line 161: Following the points above about this being a general study, not need to reference Cambrian daylength in the caption.

Point taken and both the description and reference are deleted (albeit moved to its explanation in SI).

Line 182: purely personal preference, but I’d suggest “day” and “night” rather than “daytime” and “nighttime” here and elsewhere.

Changed throughout, and we agree the text got crisper.

Line 192-5: I appreciate the removal of “Cryogenian” and “Cambrian”, but I would also avoid using “icehouse” and “greenhouse” here. I would suggest sticking to cold/cool/warm/hot as this could also reflect latitudinally different temperatures. The very generally applicable modelling work could be used to infer something about e.g. different locations in the modern ocean, like the difference between a polar setting with a steep bathymetric gradient and tropical setting with a wide shallow sea.

Fair enough. After the change, I can see the sentences may have a longer lifespan.

Line 199: consider a different descriptor (e.g. “hot” rather than “warm”) for the 40 C runs. There is a lot of difference between a 25 C ocean and a 40 C ocean, and one might expect major mortality at 40 C.

Yes, changed.

Fig 3: I again ask the authors why they have not included the 40 C simulations as z-t-O₂ space plots?

Certainly, and we discussed it internally over the review process. First, we have felt that the hot scenario is pushing the limits on expected, realistic surface ocean temperatures and may open unwanted discussion on whether the ocean could actually have surface water of +40C (and whether the clumped isotope data is reasonable etc). Although the reviewer has a fair point that this is a modeling exercise, we may also want to depict a change in fluctuations that can be imposed on an imagined sediment-water interface. Also, we evaluate that the major shift in temporal redox shifts occurs especially between +5 and +25 degrees, where we also replicate the presumed increase in TOC. If we use the higher TOC load for +40C, the model results are compared below. Although we do not think that this comparison adds information to the main manuscript, we share the model and look forward to the model’s future use ; including further exploration of other temperature-driven changes.

Fig 3E: I make one last attempt to ask the authors to clarify Fig 3E. As I said before, the y-axis scales are very different, the dashed versus solid outlines are almost impossible to differentiate, and although it's slightly more intuitive with one colour abutting each axis, it is still not explicit which axis applies to which symbol set or colour.

Yes, we see the risk for an over-filled panel. Since our point is to actually contrast these temperature different patterns, we want to avoid splitting them up into separate panels. However, we have now emphasized the symbols in different color, increased the thickness of the outline, and rephrased the caption.

Line 235-240: these sentences are purely discussion, not results, and are not needed here.
Removed.

Line 273-4: do the authors mean “growth of an individual” or “increase in population size”? Not entirely clear at present.

We agree it was not clear. We mean increase in the population size. Thank you for inviting us to clarify this and we have done so in the manuscript. The per capita growth rate captures the rate at which the population of the i th species grows per individual. So we are talking about an increase in population size, normalized by the total population size. We added an explanation that we at that specific instance refer to a species: “where the fitness-generating function, $G(v, \mathbf{u}, \mathbf{x})$, describes the expected per capita growth of a **species** as influenced by its cellular metabolism phenotype (strategy), v , the strategies of each species in the population, $\mathbf{u} = (u_1, u_2)$, and the population sizes of each species, $\mathbf{x} = (x_1, x_2)$.”.

Line 354-377: from “which could be, for example, ...” is pure discussion material. I would suggest moving all of it to the Discussion.

We thank the reviewer for this comment and do indeed agree that it was, from start, mostly discussion material. However, the last round of revision resulted in an expanded map and analysis of oxygen sensing components across animals that is not published before. To emphasize this new result, we have edited the section. We have also moved the discussion material to the discussion. What remains here is as follows:

“The population and phenotype model is agnostic to which organisms would represent pOSM or eOSM species. To relate the phenotype model, the flooding event and early animal diversification in the Cambrian Period (**Fig. 4D-E**) to the early evolution of animals, we mapped differences in the composition of HIF-driven oxygen sensing across the animal clade based on sequences in ¹⁸ and ¹⁹ (**Fig. 4F, SI text and Table S4** for details). While HIFs that today compose the OSM of modern animals contain components (e.g., genes, domains, or proteins) that can be traced back to the origins of yeast and even to prokaryotes^{42 and references therein}, its building blocks expand and differ in functions within and between early branching animals.. For example, oxygen sensing in animals with e.g., only P564 would be simpler than in animals e.g., with P564 and P402 or with P564 and CTAD. Components to the OSM like the P402, CTAD, NTAD and Factor Inhibiting HIF (FIH) regulate a wide scope of target genes at a wide range of conditions, some of them with clear importance for development⁴². From what we know of the importance of these components in Bilateria⁵⁹⁻⁶¹, the different configurations amongst animals^{18,19} reflect varying efficiency and costs to sense and respond to redox fluctuations.”

Line 419: I think “+/-“ symbol in this context is “approximately” rather than “positive 5 or negative 5 degrees C”? Suggest replacing with “~” or “about”.

We agree. The tracked addition of a ‘+’ in the previous version made it also look as if the ‘+’ sign was underscored. So it was never a “+/-“ sign and remains to be only “+5C”.

Line 465: all organisms alive now will surely “use refined mechanisms”; surely here the meaning is that

Thanks yes. This is edited for clarity to: “Eukaryotic cells (like those of animals) transport this insoluble but bio-essential ferric iron across its membrane with mechanism like the chelator ferric reductase⁶⁸.”.

Line 496-7: first clause needs referencing.

Yes, thanks. References Maxwell et al, Iwai et al., and Lando et al., are added.

Line 511: I don’t quite follow the grammar here – how does higher productivity offer large areas?

We agree this was an unclear sentence. We rephrased to clarify, as follows: “The higher productivity offers extensive habitats and biotic feedbacks where species themselves create environmental heterogeneity that, in turn, promotes the evolution and coexistence of even more species”.

Line 526: grammar of the inserted text doesn’t quite work.

We agree, now edited.

Line 533: the climate of the Ediacaran Period is still rather uncertain. Depending on which lines of evidence you follow, there may have been one long (Wang et al., 2023a, 2023b) or multiple shorter (Linnemann et al., 2022; Niu et al., 2024) glaciations in the late Ediacaran. What the climate of the early Ediacaran was like is particularly uncertain.

We thank the reviewer for pointing this out and have clarified that the timing or pattern of warming of the Ediacaran remains uncertain. Indeed, staggered events of cooling or warming may have further implications for selection of cellular mechanisms to cope with stress. The first part of the sentence now reads: “With the warm climate and enlarged continental shelves of the Ediacaran (as a shift from possibly one^{79,80} or several^{81,82} extensive glaciations) and Cambrian periods, [...]”.

Line 580-4: This is a great point and is worthy of a bit more here.

We agree and have clarified and expanded the argument.

Fig 5: something odd with the colouring of the mid-Neoproterozoic panel – blue for sky and for water?

In fact, yes. We intend for the color coding to represent cold (blue) or warm (pink) in both air and water at the icehouse versus greenhouse climate. We have clarified that in the caption.

Fig. S1: this is still quite hard to follow, though removing the red has helped. Again, I would suggest simplifying this, or making it a multi-panel plot.

We agree and have split these this SI Figure 1 up into two panels, a and b.

References cited:

- Linnemann, U., Hofmann, M., Gärtner, A., Gärtner, J., Zieger, J., Krause, R., Haenel, R., Mende, K., Ovtcharova, M., Schaltegger, U., Vickers-Rich, P., 2022. An Upper Ediacaran Glacial Period in Cadomia: the Granville tillite (Armorican Massif) – sedimentology, geochronology and provenance. *Geol. Mag.* 159, 999–1013. <https://doi.org/10.1017/S0016756821001011>
- Nardin, E., Godd ris, Y., Donnadieu, Y., Hir, G.L., Blakey, R.C., Puc at, E., Aretz, M., 2011. Modeling the early Paleozoic long-term climatic trend. *Geol. Soc. Am. Bull.* 123, 1181–1192. <https://doi.org/10.1130/B30364.1>
- Niu, Y., Shi, G.R., Zhang, Q., Jones, B.G., Wang, X., Zhao, G., 2024. Ediacaran Cordilleran-type mountain ice sheets and their erosion effects. *Earth-Sci. Rev.* 249, 104671. <https://doi.org/10.1016/j.earscirev.2023.104671>
- Scotese, C.R., Song, H., Mills, B.J.W., van der Meer, D.G., 2021. Phanerozoic paleotemperatures: The earth’s changing climate during the last 540 million years. *Earth-Sci. Rev.* 215, 103503. <https://doi.org/10.1016/j.earscirev.2021.103503>
- Valdes, P.J., Scotese, C.R., Lunt, D.J., 2020. Deep Ocean Temperatures through Time. *Clim. Past Discuss.* 1–37. <https://doi.org/10.5194/cp-2020-83>
- Wang, R., Shen, B., Lang, X., Wen, B., Mitchell, R.N., Ma, H., Yin, Z., Peng, Y., Liu, Y., Zhou, C., 2023a. A Great late Ediacaran ice age. *Natl. Sci. Rev.* nwad117. <https://doi.org/10.1093/nsr/nwad117>
- Wang, R., Yin, Z., Shen, B., 2023b. A late Ediacaran ice age: The key node in the Earth system evolution. *Earth-Sci. Rev.* 247, 104610. <https://doi.org/10.1016/j.earscirev.2023.104610>
- Wong Hearing, T.W., Pohl, A., Williams, M., Donnadieu, Y., Harvey, T.H.P., Scotese, C.R., Sepulchre, P., Franc, A., Vandenbroucke, T.R.A., 2021. Quantitative comparison of geological data and model simulations constrains early Cambrian geography and climate. *Nat. Commun.* 12, 3868. <https://doi.org/10.1038/s41467-021-24141-5>
- Wotte, T., Skovsted, C.B., Whitehouse, M.J., Kouchinsky, A., 2019. Isotopic evidence for temperate oceans during the Cambrian Explosion. *Sci. Rep.* 9, 6330. <https://doi.org/10.1038/s41598-019-42719-4>